# COMPOSITION OF MEMORY EXPERTS FOR DIFFUSION WORLD MODELS

**Sebastian Stapf & Pablo Acuaviva Huertos & Aram Davtyan & Paolo Favaro**
Computer Vision Group
Department of Computer Science
University of Bern
`{sebastian.stapf,pablo.acuavivahuertos`
`aram.davytan, paolo.favaro}@unibe.ch`

## ABSTRACT

World models aim to predict plausible futures consistent with past observations, a capability central to planning and decision-making in reinforcement learning. Yet, existing architectures face a fundamental memory trade-off: transformers preserve local detail but are bottlenecked by quadratic attention, while recurrent and state-space models scale more efficiently but compress history at the cost of fidelity. To overcome this trade-off, we suggest decoupling future–past consistency from any single architecture and instead leveraging a set of specialized experts. We introduce a diffusion-based framework that integrates heterogeneous memory models through a contrastive product-of-experts formulation. Our approach instantiates three complementary roles: a short-term memory expert that captures fine local dynamics, a long-term memory expert that stores episodic history in external diffusion weights via lightweight test-time finetuning, and a spatial long-term memory expert that enforces geometric and spatial coherence. This compositional design avoids mode collapse and scales to long contexts without incurring a quadratic cost. Across simulated and real-world benchmarks, our method improves temporal consistency, recall of past observations, and navigation performance, establishing a novel paradigm for building and operating memory-augmented diffusion world models.

## 1 INTRODUCTION

World models (WMs) are powerful tools designed to predict plausible future states of the world based on past observations Ha & Schmidhuber (2018b;a); Hafner et al. (2019); Hu et al. (2023); Bruce et al. (2024). By learning to model the distribution of the observed environment, WMs implicitly capture its underlying rules and dynamics. This capability makes them especially attractive for decision-making in downstream tasks Alonso et al. (2024); Ha & Schmidhuber (2018a); Hafner et al. (2019). Recent advances, particularly in diffusion-based world models, have demonstrated remarkable progress in generating high-quality future trajectories Hassan et al. (2024). Crucially, this predictive ability makes WMs well suited for navigation and interaction, where agents must anticipate future states in order to plan and act effectively. In such settings, an agent explores, perceives objects, and selects trajectories under uncertainty by simulating candidate futures and evaluating their outcomes before acting. These imagined rollouts are only useful if they remain consistent with prior observations; for example, a previously visited room should not spontaneously change its contents on a later visit. Maintaining such cross-time consistency is the crux of reliable prediction and decision-making.

Unfortunately, predictive WMs face a structural trade-off: richer temporal context improves fidelity but explodes compute. Transformer architectures can produce high-quality rollouts, yet their quadratic attention scales poorly and caps context length Vaswani et al. (2017). Recurrent networks and state-space models scale more gracefully in context, but compress history into hidden states, inevitably discarding detail over time. There is no silver bullet here: each family wins in one regime and loses in another. Simply "turning up" context is not a sustainable solution, because training becomes unstable and expensive, and inference costs quickly blow past practical budgets.

We advocate a different stance: memory should be distributed across a system rather than confined to a single architecture. Human cognition illustrates this principle by separating fast, capacity-limited short-term memory (STM) from slower but durable long-term memory (LTM). These forms of memory differ in both mechanism and purpose, and it is precisely this division that makes them effective together Liu et al. (2025). We adopt the same philosophy for world models (WMs): instead of burdening a single backbone with every temporal demand, we structure memory as a composition of specialized experts. This distributed approach enables WMs to balance efficiency with fidelity, supporting scalable and reliable use of past experience. Beyond composition, we address the hard reality that scaling context in training and inference is costly and brittle. Even with linear-time designs, pushing horizons indefinitely is not viable. To keep costs flat, we add a long-term memory channel that stores episodic knowledge directly in the weights of an external diffusion expert. A small number of targeted "memorization" updates amortizes future recall, enabling constant-time reuse of past experience without dragging full histories through the core model at every step.

Diffusion models are a particularly good fit for this idea because they permit principled inference-time composition of heterogeneous experts without retraining Du et al. (2023). We leverage this to integrate large pretrained backbones with lightweight adapters and auxiliary specialists into a unified framework. Naïve composition, however, is not enough: when experts still share modes that are not consistent with the past, standard product-of-experts, as we show, tends to over-sharpen common regions and collapse consistency. We therefore introduce a contrastive mechanism that factors out redundant modes during composition, preserving agreement where it matters while avoiding over-confidence where experts are merely echoing each other.

Finally, for navigation the world is not just temporal, it is spatial. Where the agent is, and how observations tie to place, is decisive for consistency. We hypothesize that anchoring memory to spatial priors (*e.g.*, pose/topological cues or map-aligned keys) improves retrieval and composition, especially in RL-style tasks that revisit locations under changing viewpoints. Accordingly, we ground our memory modules in spatial structure, so imagined futures respect both what was seen and where it was seen.

> To summarize, our **main contributions** are:
>
> 1. We aim to solve the memory problem, by using multiple expert models that excel at modeling memory at different scales and formulate it as a **product-of-experts (PoE)** problem, enabling a principled probabilistic view of memory integration in video world models;
>
> 2. We introduce **product of contrastive experts (PoCE)** a novel strategy tailored for this setting, ensuring that heterogeneous memory experts can be fused together for consistent prediction. We provide both theoretical and empirical evidence supporting its necessity;
>
> 3. We propose leveraging an **external diffusion model as long-term memory (LTM)** with a finetuning strategy that adapts pretrained priors while retaining domain-general capabilities, and extend this framework with **spatial long-term memory models** (SLTM) to further improve accuracy and spatial coherence in generated sequences.

## 2 PRIOR WORK

**Video Generation and World Models**  Recent video diffusion models have demonstrated strong capabilities in modeling temporally coherent video sequences, though ensuring long-term consistency remains an open challenge He et al. (2022); Henschel et al. (2024); Bar-Tal et al. (2024); Ma et al. (2024); Lin et al. (2024); Davtyan et al. (2023). This has motivated several lines of work, including architectural modifications such as transformer variants designed for temporal stability Tay et al. (2020); Lu et al. (2024) or different modeling frameworks Fuest et al. (2025); Ge et al. (2022). Alternative sampling strategies have also been explored, with methods that guide generation toward temporally aligned predictions based on prior context Yin et al. (2023); Chen et al. (2024); Song et al. (2025); Davtyan et al. (2023). In world modeling, early work learned dynamics from pixels; newer approaches incorporate generative memory to extend reasoning

over longer horizons Feng et al. (2023); Deng et al. (2023); Samsami et al. (2024); Alonso et al. (2024).

**Product of Experts and Compositional Adaptation** Adapting large pretrained diffusion models to new domains is challenging due to their size and domain specificity. One approach, probabilistic adaptation Yang et al. (2024a), trains a smaller diffusion model alongside a frozen pretrained one, combining their scores to flexibly adapt to new domains while retaining the robustness of the original. Combining multiple conditional experts Du et al. (2023) has been applied for image generation. Beyond images, compositionality has been explored for task and visual planning Liu et al. (2022), as well as for factorizing language instructions to guide video generation models Ajay et al. (2023).

**Memory for Video Models** SlowFast Hong et al. (2025) introduces a two-stage memory mechanism, applying LoRA adapters Hu et al. (2022) to a pretrained diffusion model for fast adaptation to recent sequences (*fast learning*), while maintaining a separate slower loop to consolidate changes for future predictions (*slow learning*). Unlike our approach, this method follows a meta-learning-style memory paradigm and requires additional training of the potentially large pretrained network. Other approaches implement memory by explicitly storing relevant past frames and learning how to retrieve and inject them into the model's context window Xiao et al. (2025). However, it introduces challenges related to increasing storage demands and retrieval complexity. Other lines of work utilize spatial memory models that retrieve relevant past context based on camera position Xiao et al. (2025); Yu et al. (2025); Wu et al. (2025). And lastly approaches that implement recurrent networks such as state-space models into their world model Savov et al. (2025); Po et al. (2025).

## 3 MEMORY ADAPTATION

### 3.1 BACKGROUND

**Denoising Diffusion Probabilistic Models** Diffusion models Ho et al. (2020) aim to learn a data distribution $q(\mathbf{x}_0)$ by introducing a sequence of $T$ progressively noisier versions $\{\mathbf{x}_t\}_{t=1}^T$ of the data $\mathbf{x}_0$. This is done via a forward Markov chain where each transition probability $q(\mathbf{x}_t|\mathbf{x}_{t-1}) = \mathcal{N}(\mathbf{x}_t; \sqrt{1-\beta_t}\,\mathbf{x}_{t-1}, \beta_t \mathbf{I})$ is Gaussian with a predefined noise schedule $\{\beta_t\}_{t=1}^T$, where $0 < \beta_t \leq 1$. The generative process learns to reverse this diffusion, modeling $p_\theta(\mathbf{x}_{t-1}|\mathbf{x}_t) = \mathcal{N}(\mathbf{x}_{t-1}; \mu_\theta(\mathbf{x}_t, t), \tilde{\beta}_t \mathbf{I})$, which approximates the reverse conditional $q(\mathbf{x}_{t-1}|\mathbf{x}_t)$. The mean $\mu_\theta(\mathbf{x}_t, t)$ is predicted using a neural network that outputs a noise estimate $\epsilon_\theta(\mathbf{x}_t, t)$. This model is trained by minimizing the simplified objective of Ho et al. (2020)

$$\mathcal{L}(\theta) = \mathbb{E}_{\mathbf{x}_0, t, \epsilon \sim \mathcal{N}(0, \mathbf{I})} \left[ \|\epsilon - \epsilon_\theta(\mathbf{x}_t, t)\|^2 \right], \tag{1}$$

which implicitly learns the score function of the data distribution $\nabla_{\mathbf{x}_t} \log p_\theta(\mathbf{x}_t) \approx -\frac{\beta_t}{\sqrt{1-\bar{\alpha}_t}} \epsilon_\theta(\mathbf{x}_t, t)$. After training, sampling iterates Langevin-style updates

$$\mathbf{x}_{t-1} = \frac{1}{\sqrt{\alpha_t}} \left( \mathbf{x}_t - \frac{\beta_t}{\sqrt{1-\bar{\alpha}_t}} \epsilon_\theta(\mathbf{x}_t, t) \right) + \sigma_t \eta, \quad \eta \sim \mathcal{N}(0, \mathbf{I}), \tag{2}$$

with $\alpha_t, \bar{\alpha}_t, \sigma_t$ defined by the noise schedule.

**Compositional Diffusion via Product of Experts** In many generative settings, it is desirable to combine the knowledge or constraints encoded by multiple models. A principled way to achieve this is through the Product of Experts (PoE) framework Hinton (2002), which defines a composite distribution as the (normalized) product of individual distributions. Each model can be viewed as an expert $p_i(\mathbf{x})$, and the query itself may define an additional constraint $p_{\text{query}}(\mathbf{x})$. The PoE then computes a joint distribution $p(\mathbf{x}) \propto p_{\text{query}}(\mathbf{x}) \prod_i p_i(\mathbf{x})$, sharply focusing on latent states consistent with all sources of evidence. This allows for selective, content-addressable generation, retrieving or synthesizing patterns that satisfy all contributing distributions. In the context of diffusion models, this corresponds to composing multiple pretrained models, $p_{\theta_i}(\mathbf{x}_t) \equiv p_i(\mathbf{x}_t)$ into a unified generative process Liu et al. (2022); Du et al. (2023).

### 3.2 COMPOSITION OF MEMORY EXPERTS

Our goal is to augment video world models with the ability to efficiently condition generation on specific past histories, while preserving the diversity and generalization of a pretrained prior. We

denote a video sequence of $T$ frames as $\mathbf{x} \in \mathbb{R}^{T \times 3 \times H \times W}$, with spatial resolution $H \times W$. The context $\mathcal{M}$ represents the set of all past frames, which could be of arbitrary length. In most modern video world models, $T$ is fixed and limited by design Lin et al. (2024); Ma et al. (2024); Yang et al. (2024b); Polyak et al. (2024). Our objective is to model the conditional distribution $p(\mathbf{x} \mid \mathcal{M})$ for coherent generation that reflects both recent and long-term context.

Different architectures offer trade-offs between consistency with the immediate past, long-term memory, generation quality, and inference speed. There is no universally optimal solution. Scaling transformers with longer context windows quickly incurs quadratic attention costs and often leads to unstable training Song et al. (2025). Sparsified attention or purely state-space models reduce memory usage but lose fidelity and long-range consistency Gu & Dao (2024); Behrouz et al. (2024).

Motivated by this, we propose to *compose specialized experts*, each responsible for a particular memory role, and fuse them probabilistically at sampling time. Diffusion models naturally support such composition through a PoE formulation, without requiring retraining Du et al. (2023). Concretely, we assume the past history $\mathcal{M}$ can be decomposed into multiple (possibly overlapping) subsets of frames $\{c_i\}_{i=1}^K$ with $c_i \subset \mathcal{M}$ and $\cup_{i=1}^K c_i = \mathcal{M}$. We assume that a diffusion model $p_i(\mathbf{x})$ has been trained to predict future frames conditioned on the corresponding context $c_i$. To approximate $p(\mathbf{x}|\mathcal{M})$, we then use

$$p(\mathbf{x} \mid \mathcal{M}) = p(\mathbf{x} \mid c_1, \ldots, c_K) \propto \prod_{i=1}^K p_i(\mathbf{x}),$$

which balances contributions from all experts.

**Product of Contrastive Experts.** The PoE framework amplifies regions where all experts assign high likelihood and suppresses others. While this sharpens consensus, in practice it can also lead to over-confident distributions and reduced diversity, particularly as the number of experts increases. To address this, we introduce a Product of Contrastive Experts (PoCM) formulation that explicitly factors out spurious distribution modes.

Formally, $\{p_i(\mathbf{x})\}_{i=1}^K$ denotes a set of heterogeneous experts. We define their composition as

$$p(\mathbf{x} \mid \mathcal{M}) \propto \prod_{i=1}^K \tilde{p}_i(\mathbf{x}), \quad \text{where} \quad \tilde{p}_i(\mathbf{x}) \propto p_i(\mathbf{x})^{\alpha_i} \, \overline{p}_i(\mathbf{x})^{1-\alpha_i}, \tag{3}$$

and where $\alpha_i > 1$ are contrasting coefficients, and $\overline{p}_i(\mathbf{x})$ denotes the expert's unconditional baseline distribution (*i.e.*, without conditioning on the context $c_i$), and $\tilde{p}_i$ are the contrastive experts. The contrastive product is motivated by the fact that each expert is only an approximation of a true distribution, and, as we show below, PoCE provides a way to suppress undesired spurious modes.

**Taming spurious modes.** A first impulse to suppress spurious modes of an approximate expert $p_i$ is to *temper* it, *i.e.*, use $p_i(\mathbf{x})^{\alpha}$ with $\alpha > 1$. This indeed downweights lighter modes more than heavier ones. However, tempering alters not only the *magnitude* of the modes but also their *spread*: for standard kernels (*e.g.*, Gaussians),

$$\left[\mathcal{N}(\mathbf{x}; \mu, \Sigma)\right]^{\alpha} \propto \mathcal{N}\left(\mathbf{x}; \mu, \tfrac{1}{\alpha}\Sigma\right),$$

so the variance shrinks by a factor $1/\alpha$. Thus, naive tempering narrows kernels and changes local geometry, which reduces diversity when sampling from it.

**Contrastive experts.** To suppress spurious modes while preserving local shapes, we combine each conditional expert $p_i$ with its unconditional baseline $\overline{p}_i$ via *contrastive mixture*: for $\alpha_i > 1$,

$$\tilde{p}_i(\mathbf{x}) \propto p_i(\mathbf{x})^{\alpha_i} \, \overline{p}_i(\mathbf{x})^{1-\alpha_i}.$$

This operation is log-linear (additive in log density), but, unlike $p_i^{\alpha}$, does not, in the KDE setting below, tighten kernels; it reweights the existing components and leaves their shapes intact.

**Proposition 1** (KDE reweighting under disjoint support). *Assume a KDE for* $p_i(\mathbf{x}) = \sum_{k=1}^M \pi_k^i h_k(\mathbf{x})$ *and* $\overline{p}_i(\mathbf{x}) = \sum_{k=1}^M \omega_k^i h_k(\mathbf{x})$ *with* $\sum_k \pi_k^i = 1$, $\pi_k^i \geq 0$, $\sum_k \omega_k^i = 1$, $\omega_k^i \geq 0$, *and* $\int h_k(\mathbf{x}) \, d\mathbf{x} = 1$.

*Suppose each kernel $h_k$ has limited bandwidth and the supports of $\{h_k\}_{k=1}^M$ are disjoint. Then the contrastive expert satisfies*

$$\tilde{p}_i(\mathbf{x}) \;\propto\; \sum_{k=1}^M \left(\pi_k^i\right)^{\alpha_i} \left(\omega_k^i\right)^{1-\alpha_i} h_k(\mathbf{x}).$$

*In particular, if $\omega_k^i = \frac{1}{M}$ (uniform baseline), the factor $(\omega_k^i)^{1-\alpha_i}$ is constant in $k$ and is absorbed into normalization, yielding $\tilde{p}_i(\mathbf{x}) \propto \sum_k (\pi_k^i)^{\alpha_i} h_k(\mathbf{x})$.*

*Proof.* On the support of $h_k$, the disjointness assumption implies $p_i(\mathbf{x}) \approx \pi_k^i h_k(\mathbf{x})$ and $\overline{p}_i(\mathbf{x}) \approx \omega_k^i h_k(\mathbf{x})$. Hence $\tilde{p}_i(\mathbf{x}) \propto (\pi_k^i)^{\alpha_i}(\omega_k^i)^{1-\alpha_i} h_k(\mathbf{x})^{\alpha_i+1-\alpha_i} = (\pi_k^i)^{\alpha_i}(\omega_k^i)^{1-\alpha_i} h_k(\mathbf{x})$. Summing over $k$ and renormalizing gives the claim. $\qquad\square$

**Interpretation and trade-offs.** By Proposition 1, the transformation $p_i \mapsto \tilde{p}_i$ leaves the *kernels* $h_k$ untouched and acts only on the *mixture weights*, via $\pi_k^i \mapsto \tilde{\pi}_k^i \propto (\pi_k^i)^{\alpha_i}(\omega_k^i)^{1-\alpha_i}$. Hence lighter modes are disproportionately downweighted when $\alpha_i > 1$. This suppresses spurious modes without distorting local shapes. The downside is that genuinely valid but *secondary* modes (with smaller weights) are also reduced, which may lower diversity; $\alpha_i$ must therefore balance *artifact removal* vs. *diversity preservation*. The KDE modeling assumptions on $p_i$ and $\overline{p}_i$ are also valid in very general settings. One can always approximate both distributions with the same basis functions (non-overlapping kernels) but different coefficients. As further discussed in Sec. A, the main requirement is instead that $p_i$ has larger magnitudes for the valid modes than for the spurious ones compared to $\overline{p}_i$, which should be a natural outcome of the training of these experts. Additional analysis and a simplified illustrative example are provided in Sec. A.

### 3.3 INSTANTIATING MULTIPLE MEMORY EXPERTS

We instantiate three complementary experts that address different aspects of temporal conditioning. Together, they form a compositional memory system that balances fidelity to recent frames, long-term consistency, and spatial disambiguation. In practice, we build upon pretrained diffusion models that generate videos from a small number of input frames. For our first memory expert, which we denote as $p_\theta(\mathbf{x} \mid c)$, we use a standard image-to-video (I2V) diffusion model. It excels at producing high-quality frames conditioned on the immediate past $c$ (in the range of 1-3 images).

**Short-Term Memory (STM).** The STM expert is a diffusion model that *directly attends* to a short-term recent context window $c_{\mathrm{ST}}$ (in the range of 10-100 images). This expert is effective at capturing local dynamics and fine details, though its temporal consistency is limited by the finite context size. We denote it as $p_\phi(\mathbf{x} \mid c_{\mathrm{ST}})$, implemented using an architecture with moderately extended context length, motivated by fast weight learning Schlag et al. (2021). The contrastive expert is the unconditional model $\overline{p}_\phi(\mathbf{x}) = p_\phi(\mathbf{x} \mid \varnothing)$. The notation $\varnothing$ indicates the absence of conditioning.

**Long-Term Memory (LTM).** While STM effectively models a limited history, many applications require consistency across hundreds of frames. Scaling the context length of existing models quickly becomes computationally prohibitive, motivating a different strategy. We therefore introduce a separate diffusion model that can be conditioned through two channels: 1) by *stores episodic knowledge in its weights* $\psi$ via test-time finetuning on the long-term history $c_{\mathrm{LT}}$ (in the range of 100-1000 images) and 2) through the standard context conditioning. To clarify the notation, let $p_{\psi(c')}(\mathbf{x} \mid c)$ denote a base diffusion model conditioned on a context $c$ and with weights conditioned on a separate context $c'$. Using $\varnothing$ as one of the contexts indicates that no conditioning has been provided or used to finetune the weights $\psi$. We then introduce $p_{\psi(c_{\mathrm{LT}})}(\mathbf{x} \mid c)$, where the parameters $\psi(c_{\mathrm{LT}})$ incorporate long-term context through finetuning on $c_{\mathrm{LT}}$ using the diffusion loss (eq. (1)). The contrastive expert is then the model without any context $\overline{p}_\psi(\mathbf{x}) = p_{\psi(\varnothing)}(\mathbf{x} \mid \varnothing)$, where $p_{\psi(\varnothing)} = p_\psi$.

A key challenge is catastrophic forgetting in the adapted model. While explicit regularization strategies such as KL penalties between base and finetuned models are possible, we find that finetuning a set of LoRA adapters Hu et al. (2022) provides sufficient implicit regularization. This approach avoids overfitting to the past, reduces computational cost, and preserves the generalization capacity of the original model.

**Spatial Long-Term Memory (SLTM).** LTM can be viewed as an associative memory that retrieves long-range knowledge conditioned on immediate visual cues $c$ Schaeffer et al. (2024). However, when cues are ambiguous (*e.g.*, looping paths or visually similar but distinct locations), additional spatial priors are essential.

To address this, we extend the context $\mathcal{M}$ with auxiliary spatial signals $\mathcal{S}$ (*e.g.*, camera poses or point maps) extracted from the history $\mathcal{M}$. We define a spatial prior $p_\lambda(\mathbf{x} \mid \mathcal{S})$, where $\mathcal{S} = \mathrm{Enc}_\lambda(\mathcal{M})$ encodes the memory into a spatial representation. Here, $\mathrm{Enc}_\lambda(\cdot)$ may correspond to a structure-from-motion algorithm (*e.g.*, SLAM Smith et al. (1986)) or a learned encoder Wang et al. (2025). This spatial prior improves long-range consistency by disambiguating locations and reducing drift in repetitive or visually ambiguous environments. The contrastive expert, as in STM, is obtained by dropping the spatial prior: $p_\lambda(\mathbf{x} \mid \varnothing)$.

**Composition of Memory Experts.** We now combine the individual experts into a unified distribution. Specializing eq. (3) to the STM $p_\phi$, LTM $p_\psi$, and SLTM $p_\lambda$, together with the pretrained prior $p_\theta$, we obtain

$$p_{\mathrm{CoME}}(\mathbf{x} \mid c, c_{\mathrm{ST}}, c_{\mathrm{LT}}, \mathcal{S}) \; \propto \; \left[ p_\theta(\mathbf{x} \mid \varnothing)^{1-\alpha_0} p_\theta(\mathbf{x} \mid c)^{\alpha_0} \right] \left[ p_\phi(\mathbf{x} \mid \varnothing)^{1-\alpha_1} p_\phi(\mathbf{x} \mid c_{\mathrm{ST}})^{\alpha_1} \right]$$
$$\times \left[ p_{\psi(\varnothing)}(\mathbf{x} \mid c)^{1-\alpha_2} p_{\psi(c_{\mathrm{LT}})}(\mathbf{x} \mid c)^{\alpha_2} \right] \left[ p_\lambda(\mathbf{x} \mid \varnothing)^{1-\alpha_3} p_\lambda(\mathbf{x} \mid \mathcal{S})^{\alpha_3} \right]. \quad (4)$$

Equation (4) shows the product-of-experts formulation where each memory model is a contrastive expert. The coefficients $\alpha_i \geq 1$ balance unconditional and conditional contributions, ensuring that recent context, long-term consistency, and spatial priors are integrated in a principled manner. For the integration of this approach into diffusion models, we refer to Sec. B.

## 4 EXPERIMENTS

In this section, we evaluate our proposed method through comparisons with baselines on synthetic and real-world datasets. We describe the datasets, baseline models, and evaluation metrics before presenting quantitative and qualitative results. Additional details, including implementation settings, computational cost analysis, extended ablation studies, and further visualizations, are provided in the appendix.

**Datasets.** We assess Composition of Memory Experts (CoME) on synthetic and real-world datasets. *Memory Maze* Pasukonis et al. (2022) consists of 30k offline trajectories (1k frames each) of agents navigating 3D mazes, with absolute and relative position signals, making it suitable for evaluating long-term memory learning. *RE10K* Zhou et al. (2018) provides indoor scenes with camera pose annotations and serves as a challenging real-world benchmark. *RECON* Shah et al. (2021) contains over 5k trajectories of real-world outdoor navigation and provides absolute camera trajectories, making it suitable for navigation planning tasks.

**Evaluation Metrics.** For high-fidelity prediction tasks (*i.e.*, low uncertainty), we report frame-wise Learned Perceptual Image Patch Similarity (LPIPS) Zhang et al. (2018), Structural Similarity Index (SSIM), and Peak Signal-to-Noise Ratio (PSNR). For navigation planning, we evaluate trajectory alignment using Absolute Trajectory Error (ATE) and Relative Pose Error (RPE), where ATE measures the global deviation of the predicted trajectory from ground truth and RPE captures the local consistency of relative motion.

### 4.1 CONSISTENCY WITH PAST OBSERVED FRAMES

**Baseline Models.** Our primary baseline is a Diffusion Transformer (DiT) Peebles & Xie (2022b), also following prior work Chen et al. (2024); Song et al. (2025). The model uses three context frames with relative positional information to predict the next seventeen frames.

We further evaluate specialized memory architectures. The short-term memory (STM) expert is a DiT(-S) with sliding window attention chunk size of 20, with 2 full attention layers at the 2 and 6 layers, taking 33 frames as conditioning and predicting the next 17 frames. The long-term memory (LTM) module adapts a pretrained DiT with LoRA finetuning. The spatial long-term memory (SLTM) variant doubles the patch size in the input projection and conditions on absolute camera poses. To

Table 1: Comparison of different memory configurations on reconstruction metrics with two steps per frame. Best values are highlighted.

| Method | LPIPS ↓ | SSIM ↑ | PSNR ↑ |
|---|---|---|---|
| Base | 0.209 | 0.771 | 19.16 |
| + STM | 0.156 | 0.820 | 21.29 |
| + LTM | 0.171 | 0.805 | 19.98 |
| + SLTM | 0.150 | 0.833 | 20.65 |
| + STM+LTM | 0.114 | 0.862 | 22.32 |
| CoME | 0.097 | 0.892 | 23.07 |
| Sliding | 0.183 | 0.753 | 19.02 |
| SSM | 0.158 | 0.828 | 20.62 |
| Full | 0.113 | 0.859 | 22.78 |

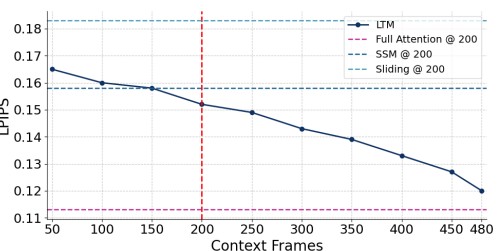

Figure 1: (LPIPS ↓) as a function of context length on the Memory Maze dataset. Longer contexts lead to more faithful reconstructions, with no saturation observed up to 480 frames.

align relative trajectories with absolute coordinates, we introduce an auxiliary encoder that maps relative positions and current clean frame estimates into absolute camera positions.

Although our approach in principle allows training-free composition of different pretrained models, we still compare with specialized architectures that represent memory-oriented alternatives: (i) a full-attention transformer as the absolute baseline, (ii) chunked transformer inference with interleaved Mamba blocks Gu & Dao (2024) for SSM-style memory, and (iii) sliding-window attention with interleaved full-attention blocks. All baselines have a comparable parameter budget and are trained for 150k steps.

We report quantitative results across SSIM, PSNR, and LPIPS (Table 1) on Memory Maze. Both LTM and SLTM are finetuned with two gradient steps per context frame. Each additional memory component consistently improves temporal consistency over the base model. In particular, augmenting with long-term memory, and especially combining STM and LTM, yields significant gains in consistency, beating all provided baselines.

Training cost is also noteworthy. Full-attention training required roughly $60\times$ the compute of STM and $12\times$ that of the base model, with also more than $3\times$ the amount of training steps to converge with the same base settings, see Section E.1. Moreover, full attention with the standard diffusion regime was unstable, requiring a modified training scheme for convergence Song et al. (2025). In contrast, our memory composition approach achieves comparable or better consistency at a fraction of the training and inference cost, demonstrating the efficiency of leveraging specialized memory experts over brute-force scaling.

**Effect of Context Length.** We further investigate the effectiveness of LTM by varying the number of context frames used for adaptation, ranging from $50$ to $480$ with 3 update steps per frame. Increasing the context length consistently improves perceptual quality, as measured by LPIPS (Figure 1). Longer contexts allow the model to leverage more temporal information and generate reconstructions that are more faithful to the ground truth. Notably, consistency continues to improve up to $500$ frames, with no saturation observed (limited by dataset length). These findings highlight the complementary roles of STM and LTM in balancing short- and long-range temporal dependencies.

**Effect on Planning.** Consistency is particularly critical for planning tasks. As a testbed for visual planning, we adopt the NavigationWM framework Bar et al. (2024), where the model is applied to goal-conditioned navigation. Given past observations and a goal image, candidate sequences of trajectories are scored by executing NWM rollouts of length eight and computing the LPIPS distance between the final frame and the goal image, averaged over three runs. Our STM is implemented as a finetuned DiT-B on twelve context frames, while the LTM is instantiated by adapting a frozen pretrained world model with LoRA finetuning.

As shown in Table 2, adding memory components consistently improves planning accuracy. Even lightweight additions such as a smaller STM yield tangible gains, supporting the principle that structured memory enhances temporal consistency and thereby benefits downstream tasks such as navigation.

Table 2: Planning accuracy on the RECON benchmark (100 sampled trajectories). We report Absolute Trajectory Error (ATE) and Relative Pose Error (RPE). Comparison of CoME with GNM Shah et al. (2022), NOMAD Sridhar et al. (2023) and NWM Bar et al. (2024).

|  | GNM | NOMAD | NWM | CoME | STM | LTM | NWM+STM | NWM+LTM |
|---|---|---|---|---|---|---|---|---|
| ATE ($\downarrow$) | 1.87 | 1.93 | 1.13 | 0.96 | 1.05 | 1.10 | 0.98 | 1.07 |
| RPE ($\downarrow$) | 0.73 | 0.52 | 0.35 | 0.28 | 0.32 | 0.32 | 0.30 | 0.33 |

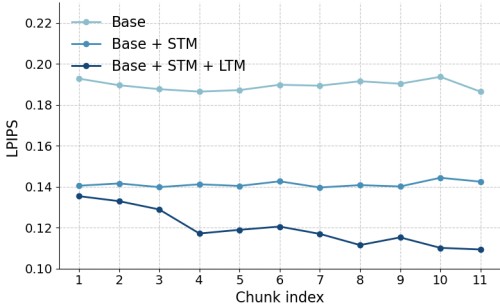

Figure 2: Evaluation on Memory Maze with 10 chunks of 20 frames each. Our method incrementally memorizes to maintain consistency.

Table 3: Comparison of evaluation metrics across different methods and rollout lengths on **RealEstate10K**. PSNR and SSIM ($\uparrow$) indicate higher is better; LPIPS ($\downarrow$) lower is better. Highlighting marks the best results.

| Method | LPIPS $\downarrow$ | PSNR $\uparrow$ | SSIM $\uparrow$ |
|---|---|---|---|
| Base | 0.405 | 19.8 | 0.794 |
| CoME | 0.359 | 21.3 | 0.83 |
| HG-v | 0.414 | 19.2 | 0.764 |
| HG-t | 0.400 | 19.7 | 0.788 |

## 4.2 Consistency with a Continuous Stream of Observations

**Streaming Evaluation.** In reinforcement learning settings, agents receive a continuous stream of observations and must remain consistent with them over time. To evaluate whether our memory composition extends beyond fixed contexts at the beginning of generation, we design the following experiment: at each step, the model memorizes the last 50 frames for 25 gradient updates, predicts the next frames, and then appends the ground-truth frames back into the context. This procedure is repeated for 11 iterations. As shown in Figure 2, on Memory Maze the STM already improves consistency across iterations by reinforcing short-term dynamics. Adding the LTM yields a compounding effect: consistency continues to improve as new observations are integrated, demonstrating that the LTM successfully stores additional information in its weights and leverages it in subsequent generations. This highlights the importance of online memory adaptation for handling continuous observation streams.

**Recall Ability.** We evaluate the recall capabilities of CoME on the RE10K dataset using a large pretrained network Song et al. (2025). In this experiment, the model generates frames and memorizes them, and then follows the reversed trajectory to reconstruct previously visited states. We test sequences of 3 forward rollouts, each followed by 3 backward rollouts. To measure recall accuracy, we compare the frames of the backward rollout with the forward rollout using LPIPS for perceptual similarity. As reported in Table 3, CoME consistently outperforms baseline methods in semantic recall. In Figure 3 we provide qualitative results, where we see that CoME accurately is able to retrieve the exact frames, preserving a consistent scene view. In contrast, the baseline model without memory generates plausible but inconsistent frames, failing to recall the original state.

**Notes on Compute.** In the online setting, we perform two memorization steps per frame, resulting in approximately 100 steps in total. When using the LTM with LoRA at rank $r = 64$, this introduces about a $4\times$ compute overhead due to memorization, in addition to the overhead from the added experts. With $r = 16$, the overhead decreases to roughly a $2\times$ increase in sampling time. Further details are provided in Section F.

## 4.3 Ablation

**Contrastive Expert Composition.** We first ablate the role of the contrastive formulation in Table 4. Applying the contrastive PoE approach improves the performance of each individual memory expert

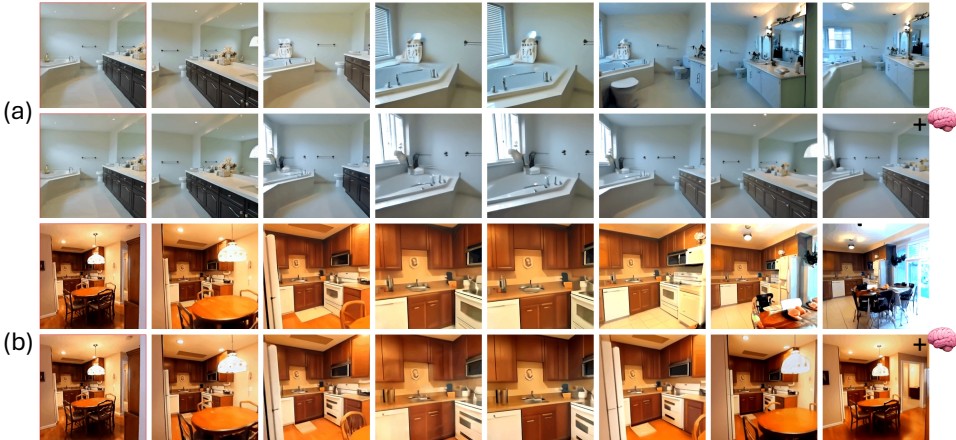

Figure 3: **Qualitative results on the RealEstate10K dataset.** We generate six forward rollouts and then reverse the camera trajectory. Unlike the base model without long-term memory, Composition of Memory Experts correctly recalls the initial frame in both examples, ensuring scene consistency. (The first frame and last frame of a sequence should be equal.)

Table 4: LPIPS results with and without the addition of the contrastive experts.

| Model | w/o Contrastive | w/ Contrastive |
|---|---|---|
| Base | 0.203 | 0.200 |
| STM | 0.175 | 0.156 |
| LTM | 0.188 | 0.171 |
| SLTM | 0.178 | 0.150 |
| STM+LTM | 0.170 | 0.114 |
| All | 0.192 | 0.097 |

Table 5: Effect of LTM adaptation capacity and context length given in LPIPS.

| | Context Frames | | | |
|---|---|---|---|---|
| Rank | 50 | 150 | 450 | Params |
| 8 | 0.193 | 0.161 | 0.146 | 1% |
| 32 | 0.175 | 0.169 | 0.128 | 4% |
| 64 | 0.161 | 0.158 | 0.124 | 7% |
| 256 | 0.162 | 0.142 | 0.118 | 25% |
| Full | 0.221 | 0.188 | 0.125 | 100% |

with the Memory Maze setting from Section 4. We can see that the naive product-of-experts quickly collapses or cancels out consistency gains, whereas our contrastive composition is essential for stable improvements. Without contrastive weighting, stacking multiple experts fails to outperform the base model and in some cases even reduces consistency.

**LTM Architecture and Context Frames.** We next analyze the effect of LTM adaptation capacity and context length in Table 5. Increasing LoRA rank consistently improves performance, while full fine-tuning tends to overfit unless the available context is highly diverse. This suggests that LoRA provides implicit regularization, enabling sufficient adaptation while retaining general priors. Longer contexts further amplify these benefits, with no saturation observed even at 450 frames, underscoring the scalability of our long-term memory design.

## 5 CONCLUSIONS

In this work, we explored how different memory mechanisms can be combined to improve the consistency and reliability of video world models. First, we demonstrated that models with complementary strengths in memory modeling can be effectively combined to yield more consistent generations. We showed analytically and experimentally that our proposed *Product of Contrastive Experts* can successfully eliminate spurious modes without introducing side-effects as in a naive Product of Experts formulation. We further introduced a novel long-term memory module that allows to store efficiently information across more than 500 frames, enabling substantially improved temporal consistency. These contributions were validated across both synthetic and real-world datasets, as well as in reinforcement learning navigation planning tasks, where reliable memory is crucial for

effective decision-making. Looking ahead, future work will explore extending long-term memory mechanisms to explicitly capture temporal dependencies, incorporating richer conditioning strategies, and developing more exhaustive forms of spatial conditioning to further enhance generative fidelity and downstream task performance.

## ACKNOWLEDGEMENTS

This work was supported by a grant from the Swiss National Supercomputing Centre (CSCS) under project ID a144 on Alps as part of the Swiss AI Initiative, and by SNSF Grant 10001278. Additional computations were carried out on UBELIX (https://www.id.unibe.ch/hpc), the high-performance computing cluster at the University of Bern.

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

## A  FURTHER ANALYSIS FOR PRODUCT OF CONTRASTIVE EXPERTS

**Non-Uniform Weights and Mode Selectivity.**  If the weights $\omega^i$ are not uniform, Proposition 1 yields

$$\tilde{\pi}_k^i \;\propto\; (\pi_k^i)^{\alpha_i} \, (\omega_k^i)^{1-\alpha_i} \;=\; (\omega_k^i) \left(\frac{\pi_k^i}{\omega_k^i}\right)^{\alpha_i},$$

where $\tilde{\pi}_k^i$ are the KDE weights for $q_i$, *i.e.*, $\tilde{p}_i(\mathbf{x}) = \sum_{k=1}^{M} \tilde{\pi}_k^i h_k(\mathbf{x})$. Let $\rho_k \doteq \pi_k^i/\omega_k^i$ denote the *contrast ratio* of conditional to baseline weights. Then $\tilde{\pi}_k^i/\omega_k^i \propto \rho_k^{\alpha_i}$, which is (strictly) increasing in $\rho_k$ for $\alpha_i > 1$. Therefore: (i) modes with $\rho_k > 1$ (more emphasized by the conditional than the baseline) are amplified relative to baseline, (ii) modes with $\rho_k < 1$ are suppressed, and the degree of separation grows with $\alpha_i$.

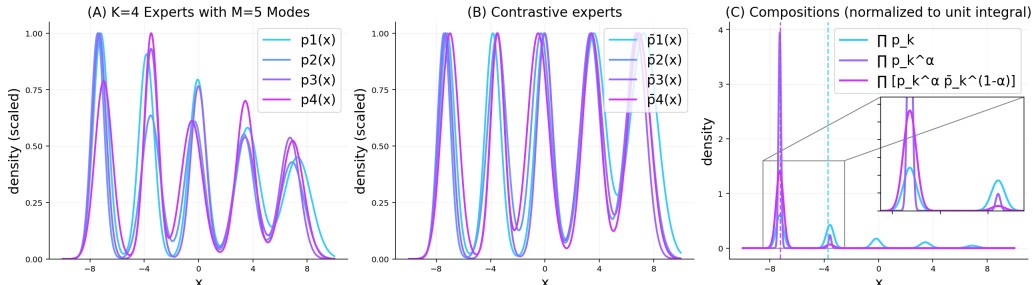

Figure 4: **Illustration of Mixture of Contrastive Experts.** (a) Individual experts, modeled as Gaussian mixtures, modes are decreasing geometrically (from left to right). (b) Individual contrastive experts, with uniform modes. (c) Product of Experts, Exponentially scales PoE, Product of Contrastive Experts.PoCE suppresses inconsistent modes (e.g., the four rightmost peaks) while preserving the dominant left kernel. The vertical line indicates the center of probability mass for the PoE and PoCE.

**Spurious vs. Secondary Modes.** To reliably remove *spurious* modes while retaining *secondary but genuine* modes, one needs a margin in the contrast ratios:

$$\exists \tau > 1 \quad \text{s.t.} \quad \rho_k \geq \tau \text{ for genuine modes} \quad \text{and} \quad \rho_k \leq \tau^{-1} \text{ for spurious modes.}$$

Under such a gap, choosing $\alpha_i$ so that $\tau^{\alpha_i}$ is large cleanly separates the two groups after renormalization. If secondary modes have $\rho_k$ only slightly above 1 (or even below), aggressive $\alpha_i$ will attenuate them too; in that case one can (a) keep $\alpha_i$ close to 1, or (b) use a baseline model with $\omega^i$s that better reflect "background" mass (*e.g.*, flatten clearly spurious regions).

**Toy Example and Visualization** To illustrate the effect of the Mixture of Contrastive Experts, we construct a toy example where each expert $p_k(x)$ is modeled as a mixture of $M$ Gaussians,

$$p_k(x) = \sum_{m=1}^{M} w_{k,m} \mathcal{N}(x \mid \mu_{k,m}, \sigma_{k,m}^2), \tag{5}$$

with weights chosen either to decay geometrically (emphasizing a few dominant modes) or to equalize peak heights across all components. The latter case defines the contrastive distributions $\bar{p}_k(x)$.

We then study their interaction by composing multiple experts using different rules:

- the *product of experts (PoE)* $\prod_k p_k(x)$, which amplifies consensus and yields various spurious likelihood modes;
- a *product of contrastive experts (PoCE)* $\prod_k p_k(x)^\alpha \bar{p}_k(x)^{1-\alpha}$, which balances experts and their contrastive counterparts.

All products are renormalized to define valid probability densities. The results highlight that standard PoE compositions concentrate probability mass around shared modes, while contrastive and interpolated forms provide control over the degree of sharpening and mitigate trivial collapse.

As shown in Figure 4, the standard PoE yields nearly equal likelihood for the two leftmost modes, centering the probability mass close to the second peak. In contrast, the contrastive composition removes the four right-hand modes almost entirely, while maintaining the structure of the dominant left kernel. This demonstrates how contrastive weighting prevents spurious consensus and yields sharper, more interpretable mixtures.

## B SAMPLING AND TEST-TIME TRAINING

**Sampling.** Given the compositional formulation in eq. (4), sampling proceeds by replacing the single-model score function in the reverse diffusion update (eq. (2)) with the joint score of all experts. At noise level $t$, we compute

$$\nabla_{\mathbf{x}_t} \log p_{\text{CoM}}(\mathbf{x}_t) = \sum_k \left[ \alpha_k \nabla_{\mathbf{x}_t} \log p_k(\mathbf{x}_t) + (1 - \alpha_k) \nabla_{\mathbf{x}_t} \log \bar{p}_k(\mathbf{x}_t) \right], \tag{6}$$

where the unconditional counterpart of each expert acts as a contrastive baseline. This additive score naturally integrates into the denoising step (eq. (2)) by substituting $\epsilon_\theta$ with the weighted combination of expert noise predictors.

**Test-Time Adaptation of LTM.** Among the experts, the Long-Term Memory (LTM) uniquely adapts online to the current episode. At the beginning of a rollout, the base diffusion prior $p_{\psi(\varnothing)}$ is finetuned on the long-term context $c_{\text{LT}}$, yielding $p_{\psi(c_{\text{LT}})}$. We perform $n_{\text{mem}}$ gradient steps on mini-batches sampled from the memory $\mathcal{M}$ using the diffusion loss (eq. (1)). To mitigate catastrophic forgetting, updates are restricted to low-rank adapters (LoRA layers), while the backbone remains frozen. This strategy injects episodic knowledge into the weights at low computational cost without degrading the pretrained prior.

After adaptation, sampling resumes with the composite score (eq. (6)). Newly observed clips are appended to the memory $\mathcal{M}$, and the LTM may be updated again for longer rollouts. This establishes a recurrent loop: the STM ensures short-term fidelity, the LTM maintains evolving long-term consistency, and the spatial prior resolves global ambiguities. Together, these components yield a stable online generation process grounded in both immediate dynamics and extended history.

## C  DATASETS

**Memory Maze** The Memory Maze environment Pasukonis et al. (2022) provides trajectories of an agent navigating through a 3D maze. However, it also provides both absolute and relative positional information of the agent within the maze. Designed for reinforcement learning, the agent's task is to navigate toward colored balls placed in the maze, with the correct color cue provided via a colored border around the frame. Relative position is represented by 2D displacement vectors and two rotation angles, while absolute position includes 2D coordinates and an orientation angle (defined with respect to the maze center). The environment supports four maze sizes. In our experiments, we use an offline dataset comprising 30,000 trajectories of 1,000 frames each, collected using a stochastic navigation policy targeting random locations under action noise. This policy ensures diverse trajectory paths that often form loops, facilitating long-term memory learning.

**RECON** The RECON Shah et al. (2021) dataset contains over 5k trajectories of real-world outdoor navigation with a Clearpath Jackal robot, providing absolute camera trajectories along with RGB, stereo, thermal, LiDAR, GPS, and IMU data, making it a comprehensive benchmark for studying latent goal models and topological memory in navigation planning.

**RealEstate10K** RealEstate10K Zhou et al. (2018) is a video dataset captured in real-world real estate environments, annotated with accurate 3D camera poses and trajectories. This rich spatial information supports training video models with explicit control over camera motion and scene geometry. The dataset is particularly valuable for memory and consistency evaluations, as it allows testing whether models can retain and recall visual information about different rooms or viewpoints within the same property. Unlike synthetic datasets, RealEstate10K provides real-world complexity and diversity, helping to validate model generalization to non-simulated settings. In our experiments, we use videos resized to $256 \times 256$.

**DeepMind Lab (DMLab-40K)** The DMLab-40K dataset consists of $64 \times 64$ resolution videos depicting agent trajectories in a 3D maze environment Yan et al. (2023); Beattie et al. (2016). The DeepMind Lab simulator procedurally generates random mazes with varied floor and wall textures. Each trajectory is 300 frames long and involves an agent traversing $7 \times 7$ mazes by selecting random target points and navigating to them using the shortest path. In total, 40,000 such action-conditioned videos are provided. The maze layout is static throughout each episode, making the environment particularly suitable for evaluating recall capabilities.

**Minecraft-200K** Minecraft-200K Yan et al. (2023) is a large-scale dataset comprising 200,000 videos of Minecraft gameplay, where the player navigates using three discrete actions: *forward*, *left*, and *right*. Each video is 300 frames long at a resolution of $128 \times 128$ pixels, with corresponding action labels for each frame. The gameplay occurs in Minecraft's marsh biome, a procedurally generated 3D world featuring complex terrain such as hills, forests, rivers, and lakes. The player

alternates between walking forward for a random number of steps and rotating randomly, causing parts of the environment to disappear and reappear in the field of view. This dynamic makes the dataset particularly well-suited for evaluating memory and temporal consistency, especially under changing viewpoints. Following the protocol of Song et al. (2025), we upsample the original frames from $128 \times 128$ to $256 \times 256$ resolution to enable higher-quality video generation.

**Memory Cards Dataset**    To provide a simplified and interpretable testbed for discrete recall, we introduce the *Memory Cards* dataset. Each sample is a $4 \times 4$ grid containing eight unique objects. The player can move up, down, left, or right, and may cover or uncover tiles. The object layout remains static during an episode. A competent world model should be able to reconstruct the complete object configuration after observing all tiles at least once. We generate 100,000 sequences of 250 frames using random actions from the `pygame` library Shinners & the Pygame community (2000–). Actions are sampled such that covering/uncovering occurs with 0.4 probability, while each directional movement (up/down/left/right) occurs with 0.15 probability, promoting frequent tile changes. The dataset is split into 90% training and 10% test sets. Additionally, we provide a specialized test set where sequences start either fully covered or uncovered and then transition through zig-zag uncovering/covering patterns.

## D    ADDITIONAL EXPERIMENTS

### D.1    MEMORY-CARDS: DISCRETE RECALL EVALUATION

This experiment evaluates the model's ability to recall individual objects that have been concealed or occluded over time. We use the discrete *Memory Cards* dataset, specifically constructed for this purpose.

We train a standard diffusion U-Net Sun et al. (2024) for 20k steps, as described in Section E.1. The evaluation input consists of video sequences showing a $4 \times 4$ grid of cards, which are sequentially revealed and concealed line by line. To measure object-level recall, we compute the tile-wise mean squared error (MSE) between the predicted and ground-truth frames at the end of each sequence. A grid tile is considered correctly reconstructed if its MSE falls below a threshold of $2 \times 10^{-5}$. Our method achieves a recall accuracy of 79.2% over a sample of 50 sequences with 20 memorization steps. In contrast, the standard diffusion (SD) baseline performs at chance level, achieving only 13%. See Section H for qualitative results.

### D.2    DMLAB: DETERMINISTIC ENVIRONMENT ANALYSIS

We replicate the Memory Maze experiment in DMLab, a fixed environment that allows for deterministic evaluation of recall capabilities. The agent initially explores the maze to collect observations, after which we generate 50 future frames and compute LPIPS scores against ground-truth frames. We use a UViT model, as described in Section E.1.

The baseline SD model achieves an LPIPS of 0.558, while our method improves this to 0.456, marking an 18% improvement. Notably, no clear overfitting ceiling is observed: increasing the number of memorization steps from 1000 to 2000 yields a modest further improvement of 0.028 in LPIPS. This suggests that, for deterministic tasks, performance may continue to benefit from additional memorization steps. Visual examples are provided in Section H.

### D.3    ABLATION: LANGEVIN CORRECTION STEPS

Motivated by techniques from Du et al. (2023) and predictor-corrector sampling in diffusion models Bradley & Nakkiran (2024), we examine whether inserting Langevin dynamics steps during sampling improves generation quality and memory recall.

We conduct this ablation on the Memory Maze dataset, using 200 context frames and generating 17 future frames. We vary both the number and scale of Langevin steps, testing fixed step sizes as well as step sizes proportional to the noise schedule $\beta_t$. Across all configurations, we observe no significant improvement over the baseline, indicating that these modifications do not provide a favorable tradeoff between computational cost and generation quality. Furthermore using unadjusted

Hamiltonian Monte Carlo, did not yield better results, with the same settings as detailed in Du et al. (2023).

### D.4 EXPERIMENT ON MINECRAFT (OASIS-STYLE SETTING).

CoME is intended as a general framework rather than a standalone world model: whenever diffusion-based world models are used, CoME can be layered on top. Beyond the diverse architectures evaluated in Tables 2–3, we additionally apply CoME to an Oasis/Diamond-style setup on the simpler Minecraft Marsh dataset. We train a diffusion model that takes 3 input context frames and predicts 17 future frames. The STM variant again uses doubled patch size for efficiency and 33 context frames to predict 17 future frames, while all other settings follow Section E.1.

Table 6: **Minecraft Marsh results (3→17 prediction).** CoME consistently improves perceptual and reconstruction metrics over the base model and single-expert variants.

| Model | LPIPS ↓ | SSIM ↑ | PSNR ↑ |
|-------|---------|--------|--------|
| Base  | 0.408   | 0.450  | 16.19  |
| +STM  | 0.375   | 0.549  | 17.03  |
| +LTM  | 0.389   | 0.552  | 17.28  |
| CoME  | 0.369   | 0.574  | 17.78  |

These results demonstrate that CoME transfers effectively to Oasis-style diffusion world models in Minecraft, yielding consistent gains across perceptual similarity (LPIPS) and reconstruction metrics (SSIM, PSNR).

## E IMPLEMENTATION DETAILS

This section outlines the experimental configurations used across all datasets to ensure reproducibility of Composition of Memory Experts. For consistency, all memorization steps, i.e., the gradient updates on past trajectories, are performed using the AdamW optimizer Loshchilov & Hutter (2019) with the same learning rate and hyperparameters as during training. Unlike prior work Hong et al. (2025), we retain the conditioning during memorization, mirroring the training setup. Regarding hyperparameters of Composition of Memory Experts, we found that simple and stable settings (e.g. the same $\alpha_i$ within a narrow, fixed range $\in [2, 3]$ for all experts) worked robustly across tasks.

### E.1 EXPERIMENTAL DETAILS

**Memory-Maze** We evaluate our approach on the Memory-Maze $9\times9$ environment using the DiT(-B) configuration Peebles & Xie (2022a). Input videos are $64 \times 64$ resolution and are subsampled every second frame. We train a VAE with $2\times$ spatial downsampling. The model takes in a total of 20 frames, 3 context frames and predicts the next 17 frames. For relative positional encoding, we concatenate translational and rotational changes and stack conditioning channels for skipped frames. In the case of relative position conditioning, we omit conditioning on skipped frames. The model is trained for 150k steps with a batch size of 16. We generate 512 videos for evaluation and exclude the ground truth context frames from metric computations.

We implement the STM as a DiT(-S) with sliding window attention chunk size of 20, with 2 full attention layers at the 2 and 6 layers, taking 33 frames as conditioning and predicting the next 17 frames. The LTM is simply the base model but with LoRA applied to query, key, value projections, MLPs, and out projections.

The SLTM is implemented as a DiT(-B) transformer with patch size 4 (instead of 2). It takes absolute 2D position and angle (2+2 dimensions) as direct conditioning. To infer absolute position from relative motion alone, we train an additional encoder composed of 12 3D convolutional layers. This encoder uses embeddings formed by combining the current prediction of clean frames with the relative positioning information. In our experiments we infuse the SLTM with the information by test-time training similar to the LTM.

Our baselines take 200 context frames and predict the next 17 frames. The full attention baseline consists of a DiT(-B) transformer with patch size 4. The SSM baseline employs a DiT(-B) transformer that processes 20 frames at a time. Every third layer integrates a Mamba Gu & Dao (2024) layer, which causally propagates information to subsequent chunks. For the sliding window baseline, we replace the Mamba global layers with full attention layers, which mimics our STM architecture.

**RealEstate10K**    For RealEstate10K, we follow Song et al. (2025), reusing the pretrained model and associated hyperparameters. All methods use 4 context frames and 4 future frames within the attention window. We compare against DFoT Song et al. (2025), evaluating both History Guidance variants: vanilla (HG-v) and temporal (HG-t). Each method generates 256 videos for evaluation, with 4 clean ground truth frames provided as context. Note that this setting differs from the interpolation setup in Song et al. (2025), so reused hyperparameters may not be optimal.

**DMLab**    For DMLab, we use a UViT backbone Hoogeboom et al. (2023), detailed in Table 7. We follow the same diffusion and sampling procedure as in Memory Maze, but employ a continuous noise schedule and operate in pixel space. Training uses a linear learning rate scheduler with 1000 warmup steps, a learning rate of $2 \times 10^{-4}$, weight decay of 0.001, and AdamW optimizer. Videos are resized to $64 \times 64$ and subsampled with a stride of 2. We train for 50k steps with a batch size of 256. Evaluation involves generating 512 videos using 50 DDIM steps. The SD baseline mirrors the sampling algorithm used for Memory Maze.

Table 7: Configuration of the UViT3D model.

| Component | Setting |
| --- | --- |
| Encoder Channels | [128, 128, 256, 256] |
| Block Types | 2×ResBlock, 2×TransformerBlock |
| Up/Down Blocks | [3, 3, 3] |
| Embedding Dimension | 1024 |
| Patch Size | 2 |
| Mid Transformer Blocks | 16 |
| Attention Heads | 4 |

**Minecraft**    We adopt the DiT architecture Peebles & Xie (2022a), diffusion framework, and training hyperparameters from Song et al. (2025). We train a latent diffusion model using precomputed latents obtained from a pretrained image VAE Rombach et al. (2021). The model attends to a 20-frame window with 3 clean context frames. For the STM, we double the patch size for efficiency and use 33 context frames to predict 17 future frames. Other settings follow Section E.1.

**Memory-Cards**    We use a UNet backbone Sun et al. (2024), detailed in Table 8. Input resolution is $48 \times 48$. The model is conditioned on 3 frames to predict 5 future frames. Diffusion and training settings follow those used for DMLab. We train for 20k steps with a batch size of 256 until convergence. For evaluation, we sample 100 videos using 20 DDIM steps.

Table 8: Configuration of the UNet3D backbone.

| Component | Setting |
| --- | --- |
| Encoder Channels | [64, 128, 256, 512] |
| Number of Resnets in Block | 8 |
| Attention Resolutions | [8, 16, 32, 64] |
| Attention Heads | 4 |

# F    COMPUTE ANALYSIS

For our compute analysis, we employ the DiT configuration used in the Memory Maze experiments, retaining most settings as described in Sections E.1. Inference with memory adaptation generally

requires up to $2 \times n - 1$ compute compared to the base model, where $n$ is the additional number of experts we chose , due to the need for multiple forward passes, one through the expert model and another through the contrastive expert. However, the actual overhead depends heavily on the size of the expert networks. In settings where the experts have significantly fewer parameters than the base model (e.g., 45% of the pretrained model's size in the STM, SLTM), the additional compute can be substantially reduced. In our Memory Maze setting, this results in only a $2\times$ increase with the addition of LTM and a $3\times$ increase in forward compute for using all experts in CoME.

Table 9: **Average runtime (in seconds) for memorization and sampling**. For 50 memorization and 50 denoising steps with different LTM, such as different LoRA ranks and a smaller adapter network.

| Adapter Config | Memorization Time | Sample Time |
|---|---|---|
| $r = 64$ | 3.33 | 1.88 |
| $r = 32$ | 2.42 | 1.88 |
| $r = 16$ | 2.07 | 1.87 |
| LTM $(45\%)$ | 4.01 | 1.38 |

We compare the memorization and sampling time in the streaming setting from Section 4, we evaluate the runtime on a Nvidia RTX 4090 by sampling 64 videos with activated TorchScript compilation. Each video involves 50 sampling steps interleaved with 50 memorization steps. Average runtimes for different adapter configurations are presented in Table 9

**Full FLOPs and parameter tables across benchmarks.** For completeness, we report the same back-of-the-envelope FLOPs and parameter analysis for all major experimental settings. Unless otherwise stated, FLOPs are computed for 50 forward denoising steps. For configurations involving test-time memorization (LTM/SLTM), we additionally report the backward-pass cost incurred during adaptation.

Table 10: **Memory Maze (DiT backbone).** Reference corresponds to 50 forward passes of the base DiT.

| Config | Total Params | Trainable | GFLOPs (fwd) | GFLOPs (bwd) |
|---|---|---|---|---|
| Base | 57,918,992 | 0 | 585.89 | – |
| STM | 32,662,720 | 0 | 206.32 | – |
| LTM | 59,500,304 | 1,581,312 | 602.83 | 419.45 |
| SLTM | 59,550,272 | 1,582,080 | 150.92 | 104.92 |
| SSM | 58,114,832 | 0 | 6,293.09 | – |
| Sliding | 57,918,992 | 0 | 11,198.53 | – |
| Full | 57,918,992 | 0 | 5,858.95 | – |
| DiT (Match) | 172,645,648 | 0 | 1,752.87 | – |

Table 11: **Navigation World Models (NWM).** Reference corresponds to 50 forward passes. CoME configurations include 25 backward passes for memorization.

| Config | Total Params | Trainable | GFLOPs (fwd) | GFLOPs (bwd) |
|---|---|---|---|---|
| CDiT-XL | 675M | 0 | 1,719.87 | – |
| CDiT-L-LTM | 464M | 6.3M | 1,182.43 | 805.95 |
| CDiT-B-STM | 231M | 0 | 1,405.33 | – |

These numbers highlight two practical points: (i) the added cost of CoME comes primarily from extra forward passes rather than from a large increase in trainable parameters (LoRA adapters are ~1.6M parameters), and (ii) scaling via heterogeneous experts can be competitive with common single-model alternatives such as sliding-window or SSM-style approaches under comparable rollout settings.

Table 12: **RealEstate10K (R10k).** Reference HG-t uses 3 classifier-free guidance (CFG) evaluations per step. CoME includes 100 backward passes for memorization.

| Config | Total Params | Trainable | GFLOPs (fwd) | GFLOPs (bwd) |
|---|---|---|---|---|
| UViT-Base | 288M | 0 | 2,420.06 | – |
| UViT-LTM | 293M | 4.6M | 2,493.46 | 1,161.93 |
| UViT-HG-v | 288M | 0 | $2 \times 2,420.06$ | – |
| UViT-HG-t | 288M | 0 | $3 \times 2,420.06$ | – |

## G LIMITATIONS

While CoME provides a flexible and effective framework for composing heterogeneous memory experts, CoME relies on the assumption that each expert provides plausible and reasonably calibrated predictions. In practice, we observed failure cases primarily when one expert produced outputs that the remaining experts considered implausible. This typically occurred when an expert's expressiveness was severely constrained (e.g., due to aggressive parameter reduction), preventing it from reproducing basic scene structure. In such cases, the fused prediction can degrade in quality. While we did not observe instability under normal settings, CoME remains sensitive to the quality and capacity of its constituent experts.

## H ADDITIONAL VISUAL EXAMPLES

This section provides qualitative results that complement the main experiments described in Section E.1. For each dataset, we visualize the generations produced by our model after receiving a fixed context window. The context frames are highlighted in red. The sampling configuration matches that used in Section E.1.

We present results across the Memory Maze (Figures 5 and 6), RealEstate10K (Figures 7 and 8), DMLab (Figure 9), and Memory Cards (Figure 10) environments, highlighting spatial and temporal coherence in the generated rollouts. For DMLab datasets, we pick videos where the memory is required, so when the actions lead to a revisit of the scene. For the other datasets, we randomly pick videos for visualization.

## I LARGE LANGUAGE MODEL USAGE

Large language models were used solely to assist in revising and improving the clarity, structure, and grammar of the camera-ready manuscript. No LLMs were used for generating experimental results, modifying data, designing algorithms, or conducting analyses. All technical content, experiments, and conclusions were developed and verified by the authors.

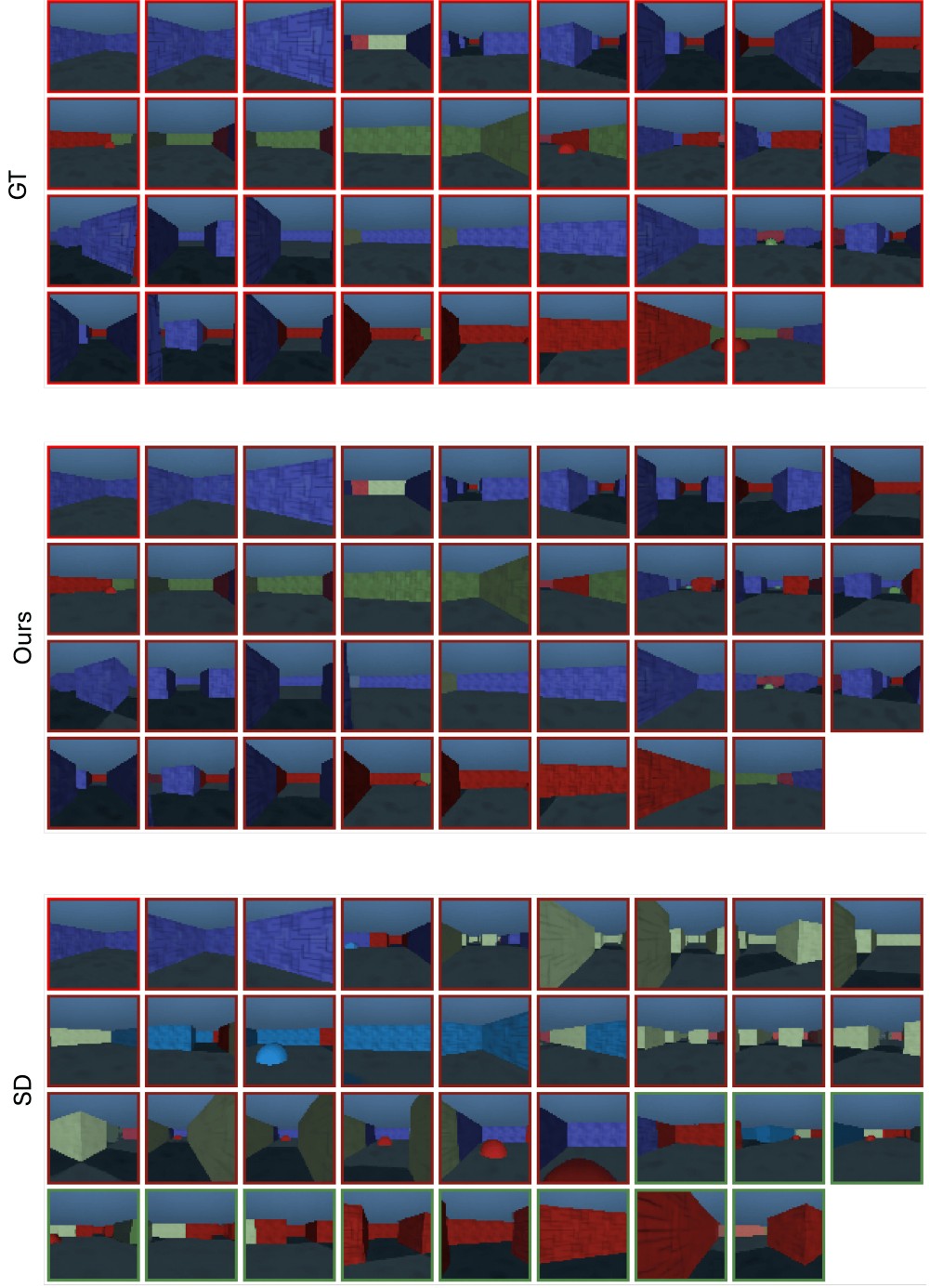

Figure 5: **Memory Maze – CoME.** After memorizing a trajectory through the maze, and only given three context frame, the model generates frames that accurately reflect the correct turns and re-entries.

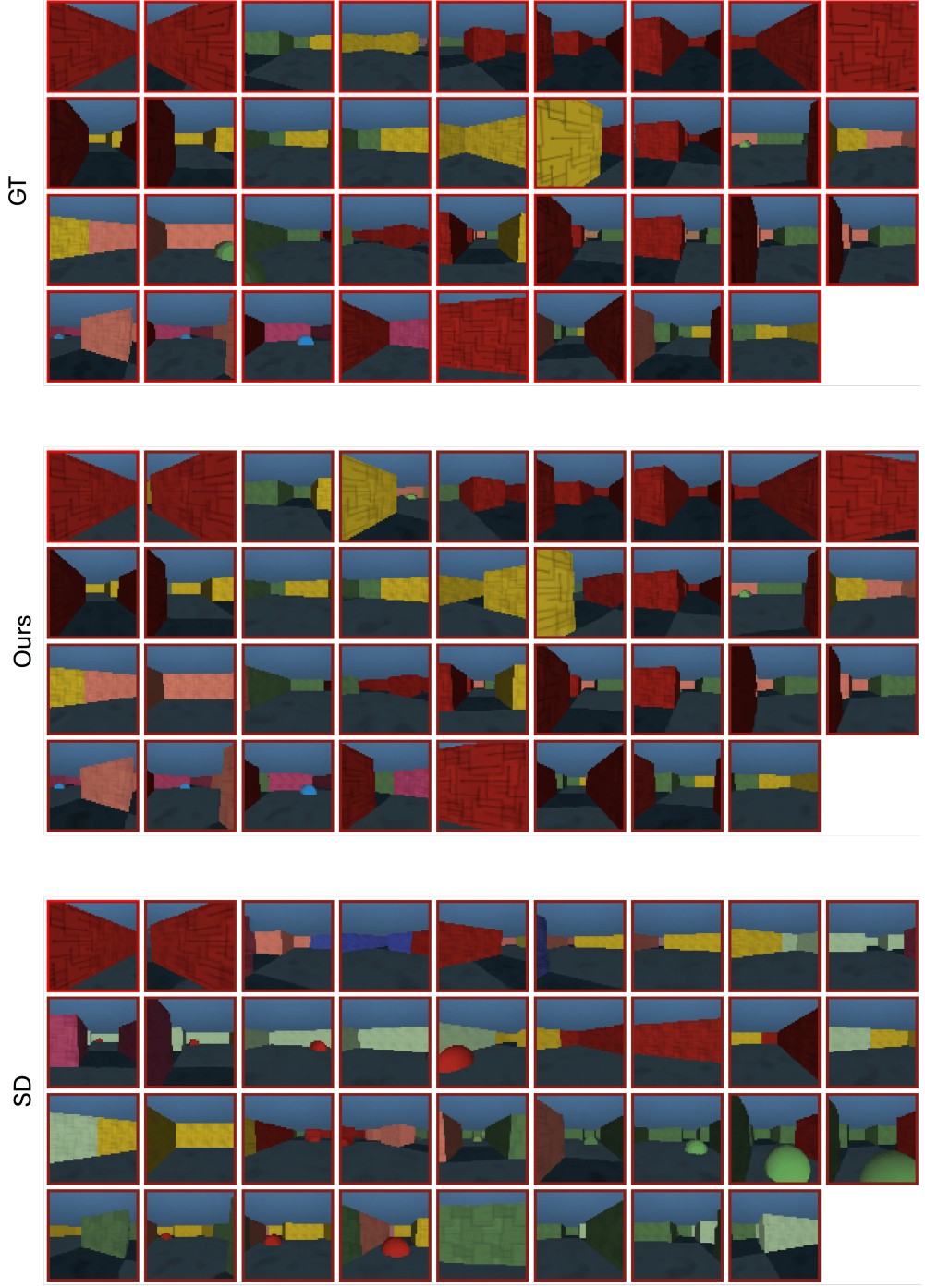

Figure 6: **Memory Maze – CoME.** After memorizing a trajectory through the maze, and only given three context frame, the model generates frames that accurately reflect the correct turns and re-entries.

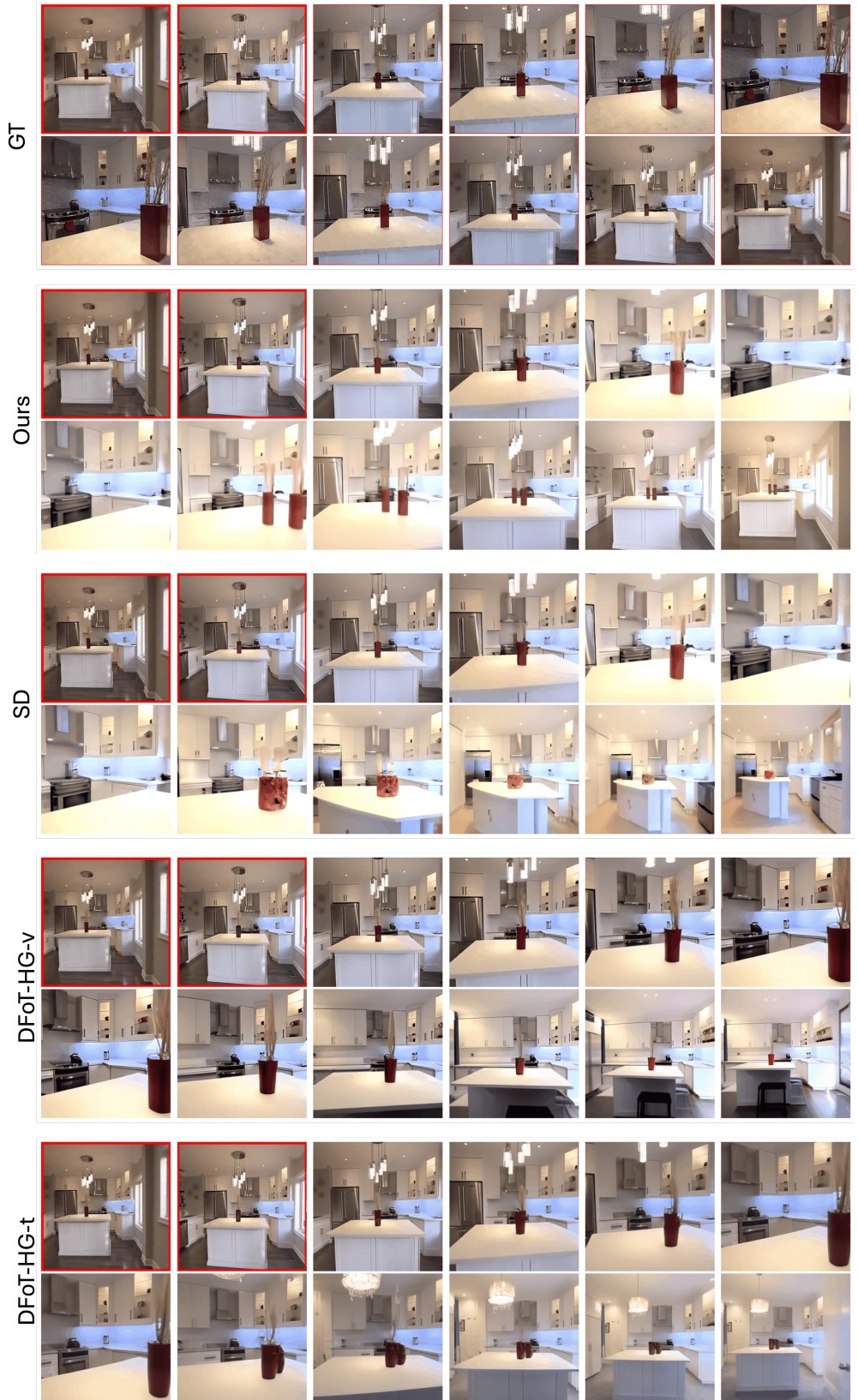

Figure 7: **RealEstate10K - CoME.** We provide 4 ground truth context frames, here highlighted with a red border. We generate 3 forward rollouts and 3 rollouts with the reversed trajectory camera position.

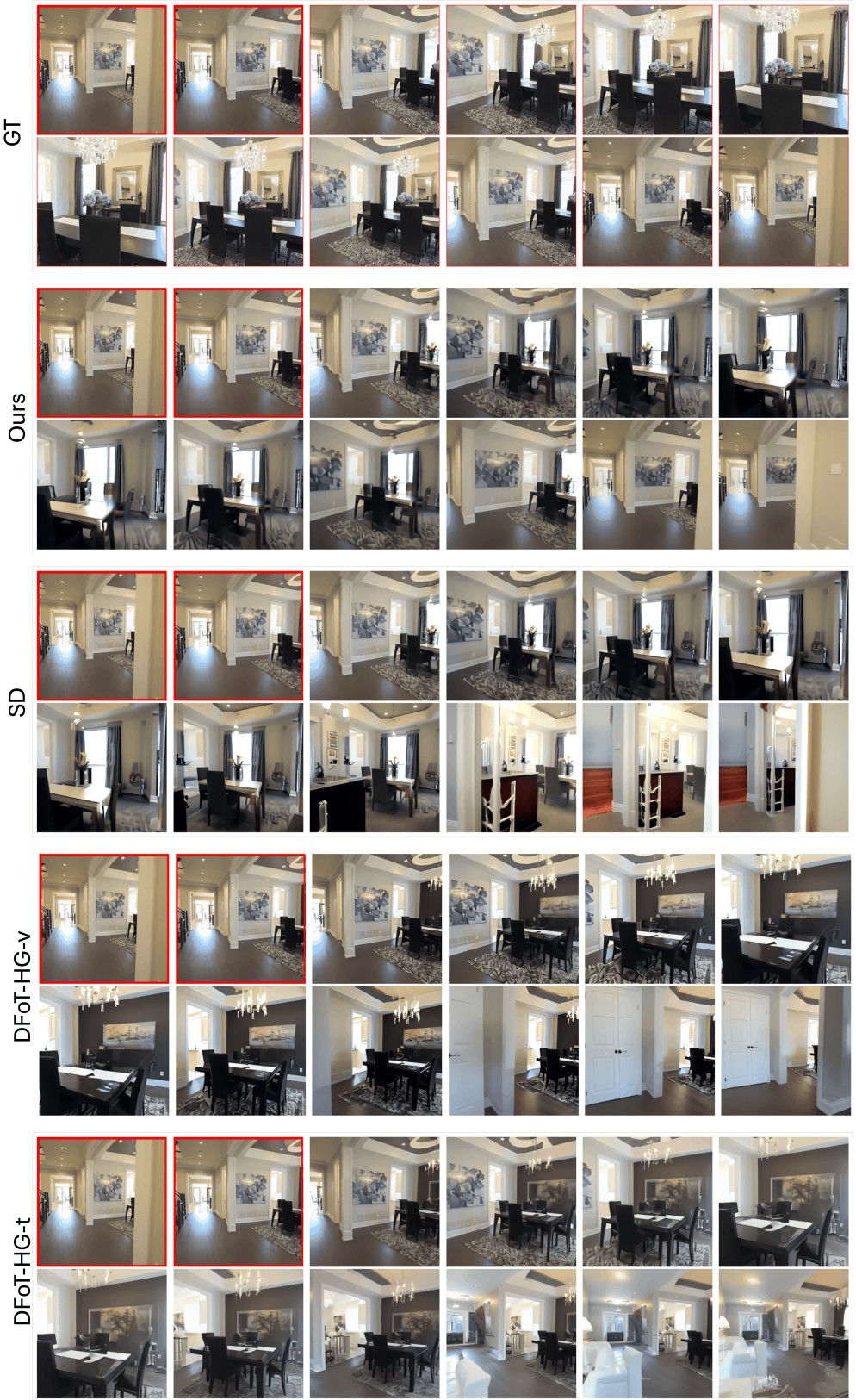

Figure 8: **RealEstate10K - CoME.** We provide 4 ground truth context frames, here highlighted with a red border. We generate 3 forward rollouts and 3 rollouts with the reversed trajectory camera position.

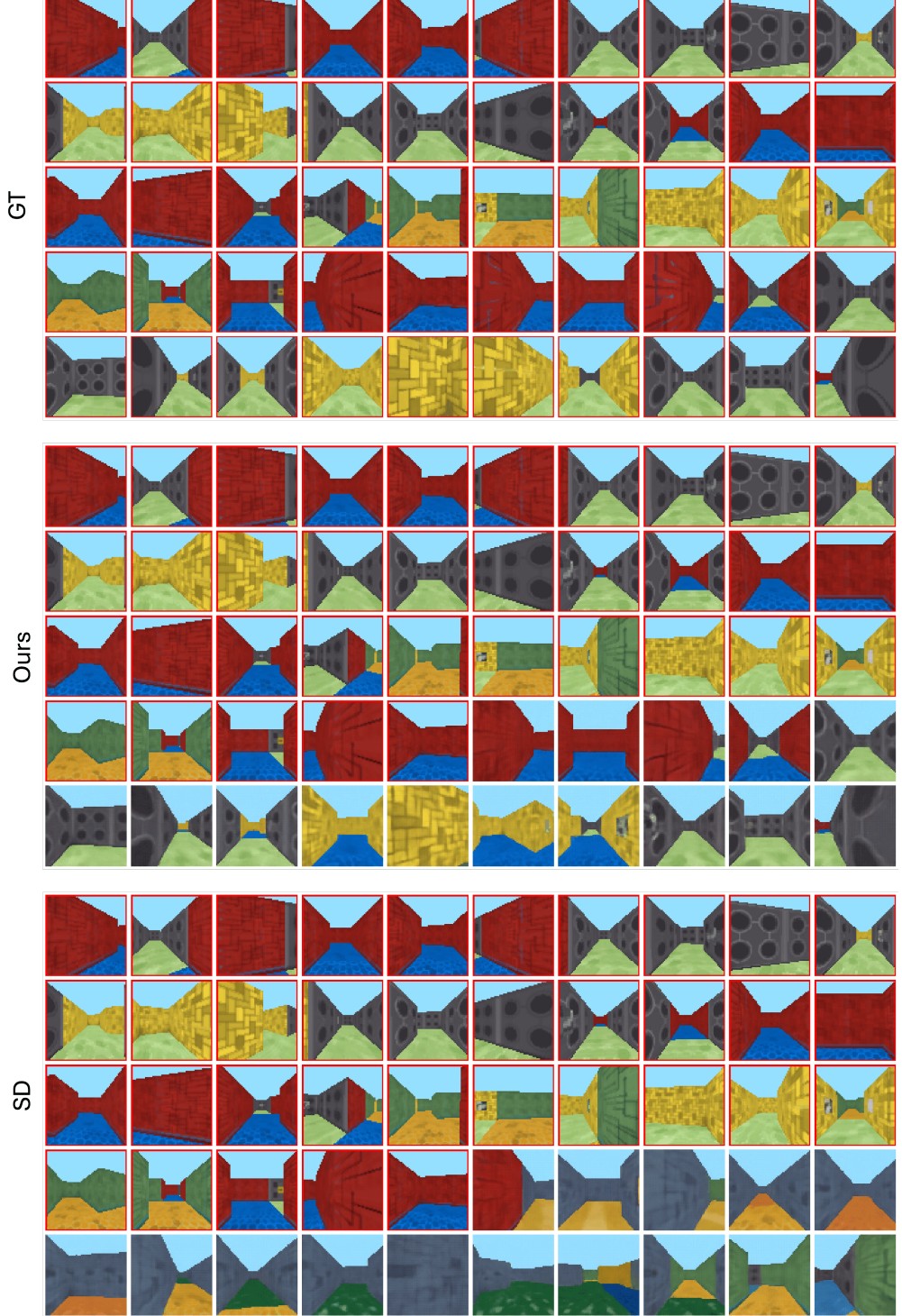

Figure 9: **DMLab - CoME.** Given 50 context frames, here highlighted with a red border, we visualize the next 4 rollouts. Our method produces coherent forward trajectories that reflect consistent agent movement and scene structure.

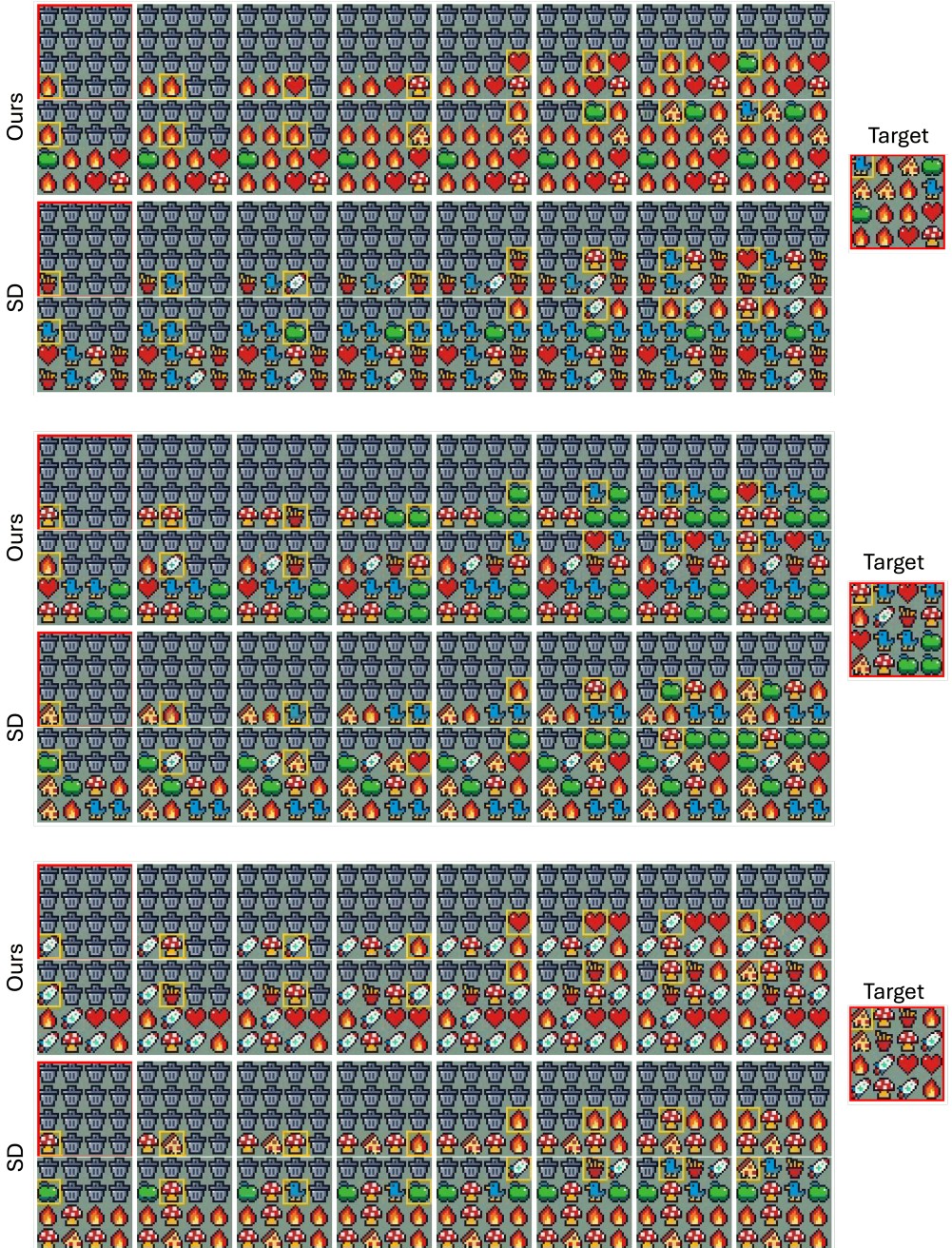

Figure 10: **Memory Cards — Discrete object recall.** At the end of the sequence, the model is evaluated on its ability to regenerate occluded tiles. After being shown a sequence of uncovering and covering actions, such that all tiles were visible at some point, our method more accurately recalls the occluded tiles, here given as 'target', demonstrating effective memory recall.

