# OpenReview forum: "Composition of Memory Experts for Diffusion World Models"
_ICLR.cc/2026/Conference — ICLR 2026 Poster_

### Official Review · Reviewer_hHXL · 2025-10-23

**Soundness:** 3
**Presentation:** 4
**Contribution:** 4
**Rating:** 6
**Confidence:** 3

**Summary:**

This paper proposes a memory-compositional diffusion world model that integrates multiple specialized “memory experts”—short-term memory (STM), long-term LoRA-based memory (LTM), and spatial long-term memory (SLTM)—into a unified model via a **Product of Contrastive Experts (PoCE)** mechanism. The PoCE combines multiple memory outputs while reducing redundant features through contrastive weighting, encouraging each expert to specialize. Experiments on navigation and reconstruction datasets show clear improvements in long-horizon consistency, memory recall, and generalization to novel sequences.

**Strengths:**

* The conceptual idea of composing multiple memory experts through a contrastive product is innovative and well-articulated.

* The use of test-time LoRA as long-term memory is practical, allowing scalable specialization without retraining the core model.

* Experiments convincingly demonstrate that the composition improves both reconstruction quality and navigation performance.

* The paper is clearly written, with mathematical formulation and visual illustrations that make the mechanism understandable.

* The approach connects well to emerging work on compositional generative models and memory-based reasoning.

**Weaknesses:**

* The assumptions behind the contrastive composition (such as the independence of expert outputs) are not empirically validated.

* Computational overhead and runtime comparisons to single-memory baselines are missing.

* While results are strong, the evaluation scope is somewhat limited to a few datasets.

**Questions:**

1. How sensitive is performance to the number of experts or their relative weights?

2. Could you provide compute and runtime comparisons to conventional diffusion world models?

3. Are there stability issues when experts disagree strongly during sampling?

---

> ### Author Response · Authors · 2025-11-16
>
> We thank the reviewer for the thoughtful feedback and address each concern in detail below.
>
> ## Weaknesses
>
> **Assumptions behind contrastive composition.**
> The reviewer points out that the independence assumptions in our contrastive composition are not empirically verified. In our framework, this independence serves as a modeling simplification rather than a literal statistical claim. Each expert defines an energy or constraint over x (eq.3), and their product emphasizes regions where their preferences align. Thus, “independence” here refers to treating the experts’ evaluations as conditionally independent given the input, not to independence of random variables.
> This assumption follows the standard product-of-experts paradigm. In our setting, it is also empirically supported: as shown in Table 4, enforcing this factorization consistently improves sample quality across tasks. These gains depend directly on this modeling structure and would not emerge without it.
>
>
> **Computational overhead and runtime comparison.**
> We now include full compute and runtime comparisons to the baselines. These results appear directly below and in the responde to reviwer *LCCy* in more detail ("Full FLOPs / Parameter Tables"). Moreover, we point to the additional results in  Section 4.2 and Section F., where we provide an auxiliary runtime analysis.
>
> **Evaluation scope.**
> The reviewer suggests that our evaluation is limited in scope. We disagree here. Our experiments span five datasets that collectively cover visual prediction, planning, and memory-intensive world-modeling scenarios, across real and synthetic worlds. This diversity is essential for evaluating methods that must perform consistently across different forms of temporal and spatial uncertainty. CoME exhibits strong improvements across all of them, underscoring the generality of the approach.
>
> ## Questions
>
> **Q1. Sensitivity to number of experts and expert weights.**
> Our ablations (Table 1) show a clear trend: additional experts consistently improve performance. A modest hyperparameter search further revealed that the expert weights $\alpha_i$​ yield similar performance by fixing them between the range $[2,3]$. We therefore found CoME not to be particularly sensitive to fine-tuning these values. We will make sure to add these specifications and results to the final version.
>
> **Q2. Compute/runtime comparisons to diffusion world models.**
> See above for compute and runtime comparison.
>
> **Q3. Stability under strong expert disagreement.**
> Strong disagreement between experts is expected in many of our experiments, especially when the base model lacks information available to the memory experts (e.g., occluded regions in Memory Maze). In such cases, the base model captures a broad set of possible modes, while the expert model focuses more confidently on specific, plausible futures. Under the product-sampling formulation, stability is maintained as long as the memory expert proposes states the other model still considers likely. This is precisely the setting CoME is designed for, and empirically, we observe no instability across tasks.
>
> Failures arise only when one expert provides information that others deem implausible. In our experiments, this occurred when an expert’s expressiveness or generation quality was severely limited, e.g. when reducing the amount of parameters to much, failing to reproduce even basic structural elements, leading to degraded overall generation quality. We will add this discussion in a limitation section in the final paper.
>
> ## Compute Analysis
>
> #### **For Table 1 – on Memory Maze**
>
> **Back-of-the-envelope calculation:**
> - *Reference:* 50 forward steps dit_full $50 *5858 = 292k\text{ GFLOPs}$
> - *CoME*: 50 forward steps:  $50 * (150 + 600 + 206 + 585)\text{ GFLOPs} *2 = 144k\text{GFLOPs}$
> 	 - $+ (100 \text{ Backward Steps}*(419+104)\text{ GFLOPs}=52k \text{ GFLOPs}$
>
> | **config** | **total**   | **trainable** | **GFLOPs(fwd)** | **GFLOPs(bwd)** |
> | ---------- | ----------- | ------------- | --------------- | --------------- |
> | Base       | 57,918,992  | 0             | 585.89          | n/a             |
> | STM        | 32,662,720  | 0             | 206.32          | n/a             |
> | LTM        | 59,500,304  | 1,581,312     | 602.83          | 419.45          |
> | SLTM       | 59,550,272  | 1,582,080     | 150.92          | 104.92          |
> |            |             |               |                 |                 |
> | SSM        | 58,114,832  | 0             | 6,293.09        | n/a             |
> | Sliding    | 57,918,992  | 0             | 11,198.53       | n/a             |
> | Full       | 57,918,992  | 0             | 5,858.95        | n/a             |

---

### Official Review · Reviewer_9rob · 2025-10-28

**Soundness:** 3
**Presentation:** 3
**Contribution:** 3
**Rating:** 8
**Confidence:** 4

**Summary:**

In this paper, the authors propose a novel diffusion world model that employs a composition of multiple types of memory experts to overcome the *memory-compute trade-off* inherent in existing architectures (e.g., Transformers, State Space Models (SSMs)). This model generates the next frame by probabilistically fusing the predicted probability distributions from each expert into a single, unified distribution.

The paper introduces three types of experts: a **Short-Term Memory (STM)** expert that predicts based on recent context, a **Long-Term Memory (LTM)** expert that stores long-term context in its weights via test-time fine-tuning, and a **Spatial Long-Term Memory (SLTM)** expert that utilizes spatial knowledge for prediction . In their experiments, the authors validate the positive impact of each memory component on model performance. They also demonstrate that performance **monotonically increases** as more context is provided to the LTM.

Furthermore, the authors propose a novel formulation, the **Product of Contrastive Experts (PoCE)**, to ensure that each expert focuses more intently on the given context . This method selectively amplifies predictions that have a higher relative probability when conditioned on the context, compared to an unconditional baseline . Through experiments, they show that this approach mitigates the **consistency collapse** phenomenon—often observed in the conventional Product of Experts (PoE) method—and achieves superior performance .

Finally, the authors experimentally validate that their full model, combining the STM, LTM, and SLTM experts with the PoCE unification formulation, outperforms existing memory-augmented world models. In the appendix, they provide a detailed description of their experimental setup and demonstrate the effectiveness of PoCE through a toy example . They also validate the superior performance of their world model through a discrete recall evaluation (Memory-Cards environment) and a deterministic environment analysis (DMLab) . Notably, in the discrete recall evaluation, their model improves recall accuracy by approximately **66%** compared to a standard diffusion baseline (79.2% vs. 13%).

**Strengths:**

- **Addresses a significant research question and proposes a viable alternative:** The authors tackle the critical _memory-compute tradeoff_  in world models, a problem where models are either limited to using short-term memory or incur excessive computational overhead when employing long-term memory . To address this, they introduce the concept of a Product of Experts (PoE), proposing a model where specialized memory experts operate compositionally, and they experimentally validate its effectiveness through significant performance improvements.

- **Novel formulation for a contrastive Product of Experts:** The authors do not simply adopt the standard PoE framework. They identify a potential issue of consistency collapse when applying it to memory experts and propose a novel contrastive formulation to mitigate this problem . The superiority of this method is convincingly demonstrated through both an intuitive toy visualization and an empirical ablation study .

- **Thorough experimental validation and detailed methodology:** The authors demonstrate the effectiveness of their proposed compositional memory experts through a comprehensive set of ablation studies. For the Long-Term Memory (LTM) expert, in particular, they show a monotonic performance improvement as the context length increases . Furthermore, they strengthen their claims by including additional insightful experiments in the appendix, such as the PoCE toy visualization and a discrete recall evaluation, which experimentally prove the model's efficacy . The paper is also well-written, with the methodology and experimental settings described in sufficient detail in both the main paper and the appendix to ensure reproducibility.

**Weaknesses:**

- **Insufficient discussion on whether the proposed method truly resolves the memory-compute tradeoff:** The authors frame their work as a solution to the memory-compute tradeoff. However, their proposed method, Composition of Memory Experts (CoME), requires forward passes for each expert and its contrastive counterpart, leading to a computational complexity increase of approximately $2\times n$ (where $n$ is the number of experts), as discussed in Appendix F. This raises doubts as to whether CoME fundamentally solves the issue, as requiring more memory (i.e., more experts) still leads to a direct increase in computation. While the authors mention in Appendix F that this overhead can be reduced by using smaller expert models, this point is critical to the paper's central research question and warrants a more in-depth discussion with concrete experimental results in the main paper, not just the appendix .

- **Lack of discussion on limitations:** The paper lacks a thorough discussion of the limitations of the CoME framework. For instance, as the number of experts with diverse characteristics increases, the computational overhead will grow, and the process of fusing their predicted distributions could become more challenging (e.g., tuning the hyperparameters $\alpha_i$ ​may become prohibitively complex). Moreover, while the framework is effective if the chosen memory experts are well-suited for the task, inappropriate experts could have a detrimental impact, potentially leading to performance worse than the base model. A discussion of these potential failure points and scalability challenges should be included.

- **Insufficient experimental analysis for Short-Term Memory and Spatial Long-Term Memory:** While the paper demonstrates a monotonic performance improvement for Long-Term Memory (LTM) with increased context, similar in-depth analyses for Short-Term Memory (STM) and Spatial Long-Term Memory (SLTM) are missing. The ablation studies confirm that using these modules is beneficial, but they do not provide clear evidence that the performance gains are achieved for the intended reasons—i.e., that STM excels by focusing on recent observations and SLTM by leveraging spatial knowledge. This makes it difficult to verify whether these components are functioning as hypothesized.

- **The Statement about the Use of Large Language Model:** The authors doesn’t state explicitly how they used LLM in their submission, which is guided in https://iclr.cc/Conferences/2026/AuthorGuide.

- **(Minor) Comments on Presentation:**
    - In line 173, explaining the "subsets of frames" with a concrete example would help prevent reader confusion about how these subsets are determined.
    - In line 221, there appears to be a small formatting artifact at the end of the line.
    - In Section 3.3, the description of SLTM is ambiguous about its use of fine-tuning. This should be clarified in the main text, as it is mentioned in Appendix E.1.
    - The discussion on training cost (lines 359-364) could be more detailed with the training time for each model.
    - In Section 4.3, the dataset used for the ablation studies should be explicitly mentioned.
    - In Figure 3, the meaning of each row and what the reader should observe (e.g., which frames are forward vs. reverse) is not clearly explained in the caption.
    - In line 975, it should be explicitly stated that the "$2\times$" increase is in comparison to the base model.
    - In Figures 5-10, the red box annotations appear misaligned or incorrectly drawn in several places (e.g., all frames are boxed in "Ours" in Figure 5; boxes are unclear in Figure 10).

**Questions:**

- Could you please share a more detailed experimental analysis for the Short-Term Memory (STM) and Spatial Long-Term Memory (SLTM) experts? For instance, it would be insightful to see an experiment where, in the absence of SLTM, the model's performance on a dedicated spatial reasoning task (e.g., as explored in [1]) is shown to decrease. This would provide stronger evidence that the SLTM is indeed contributing the specific spatial capabilities as intended.

[1] Cho, Junmo, Jaesik Yoon, and Sungjin Ahn. "Spatially-aware transformer for embodied agents." _arXiv preprint arXiv:2402.15160_ (2024).

---

> ### Author Response · Authors · 2025-11-16
>
> We thank the reviewer for the feedback. Below we address the raised concerns in detail.
>
> ## Weaknesses
>
> **Insufficient discussion on whether CoME resolves the memory–compute tradeoff**
> We agree that a deeper computational analysis would make the argument clearer. We are adding a comparison showing that combining several smaller experts with complementary strengths outperforms a single larger model under comparable, and in some cases lower, compute budgets. The results can be found in more detail in the answer to reviewer *LCCy* ("Full FLOPs / Parameter Tables"). This supports our claim that CoME allows scaling memory capacity without incurring the same compute growth seen when scaling a single monolithic model. We will integrate these results and a more detailed discussion into the main paper.
>
> **Lack of discussion on limitations**
> We observed failures primarily when one expert produced outputs the others deemed implausible. This occurs when an expert suffers from severe expressiveness bottlenecks, e.g. after aggressive parameter reduction leading to inability to reproduce basic structure. This, in turn, degrades the fused prediction. Regarding hyperparameters, we found that simple and stable settings (e.g. the same αᵢ within a narrow, fixed range  $\in[2,3]$ for all experts) worked robustly across tasks. We will include concrete examples of these failure cases and a clearer discussion of scalability limits and tuning guidelines in the final version.
>
> **Insufficient analysis of STM and SLTM**
> The benefit of SLTM is evident from the fact that it improves performance even when LTM is active. Both share similar memorization dynamics, so the additional gains can be attributed to the spatial signal. We also realized that the STM description in the main text was incomplete, possibly leading to confusion.
> In section 4.1 we write
> *"The short-term memory (STM) expert is a smaller DiT with a 20-frame sliding-window attention mechanism and two additional full-attention layers."*
>
> However Section E specifies the correct design:
> *"We implement the STM as a DiT(-S) with sliding window attention chunk size of 20, with 2 full attention layers at the 2 and 6 layers, taking 33 frames as conditioning and predicting the next 17 frames."*
>
> This architecture restricts STM strictly to short-range temporal context, ensuring it extracts only local temporal cues. We will integrate this clarification and expand the analysis to show how STM and SLTM contribute complementary benefits.
>
> **Missing statement about LLM usage**
> We will add the required statement to the final version.
>
> **Minor presentation comments**
> Thank you for these detailed notes. We will revise the affected lines.
>
> ## Questions
>
> **STM and SLTM experiment**
> We think that all of the tasks we evaluate inherently require spatial reasoning, as generation quality depends on accurately locating the agent/object relative to its surroundings and predicting occluded or future structures (e.g., determining what lies beyond a wall in Memory Maze).
>
> However, we realize that there is ambiguous writing in Section 4.1. We want to highlight that the LTM in the “Effect on Planning” subsection in Section 4.1 follows the NavigationWM framework. That is, it is actually an SLTM, since it is conditioned on absolute 3D coordinates. Because trajectory planning based on model predictive control is a task requiring spatial memory, we see the interplay of the STM and SLTM in Table 2. Thus, we believe that this experiment already covers the requested analysis. We will make sure to correct this in the camera-ready version.

---

> > ### Comment · Reviewer_9rob · 2025-11-24
> >
> > The authors have adequately addressed my concerns and questions. I appreciate the detailed responses and the additional experiments.

---

### Official Review · Reviewer_DBh7 · 2025-10-31

**Soundness:** 3
**Presentation:** 4
**Contribution:** 3
**Rating:** 8
**Confidence:** 4

**Summary:**

To leverage the benefits from different memory models, the proposed approach (CoME) utilises a diffusion framework with a contrastive mechanism to compose information from memories. Experiments across different settings showed that the strategy improves performance.

**Strengths:**

- The mechanism to compose heterogeneous experts is well-presented in Section 3.2, and different memory models (long-term, short-term, and Spatial Long-term Memory) are detailed clearly in Section 3.3.
- Experiments were conducted on a range of datasets, including Memory Maze, RealEstate10K, DeepMind Lab, Memory Cards Dataset, and RECON.  The comparisons of different memory combinations are also highlighted in Table 1. The setting in section 4.2 shows that the composed mechanism can handle a stream of observations.

**Weaknesses:**

Since the approach proposed to use all memory models, it will be more expensive than using one memory. However, the authors also discussed and provided compute analysis in Appendix F.

**Questions:**

The author could highlight more in the baseline, which one is the baseline that contains the simpler way of combining knowledge from memory models.

---

> ### Author Response · Authors · 2025-11-16
>
> We appreciate the reviewer’s time and effort put into writing this review and adress the raised concerns below.
>
> ## Weaknesses
>
> **Computational cost of leveraging all memory models** Because multiple reviewers raised this point, we have expanded our analysis in the revision. The updated paper will  include a detailed breakdown of parameter counts and FLOPs for every expert individually as well as for all expert combinations.
> These results which are found below and in more detail in our response to Reviewer LCCy ("Full FLOPs / Parameter Tables"), show the trade-offs more clearly and allow for direct comparison with single-model alternatives.
>
> ## Questions
>
> **Clearer baseline identification** Table 4 reports the comparison of CoME (w/ Contrastive) to the simpler method that combines knowledge without the product-of-contrastive-experts formulation (w/o Contrastive). We agree that Section 4.1 did not explicitly highlight this, and we will revise the text to state clearly that this is the baseline used to evaluate the benefit of our way of composing multiple experts.
>
> ## Compute Analysis
> #### **For Table 1 – on Memory Maze**
>
> **Back-of-the-envelope calculation:**
> - *Reference:* 50 forward steps dit_full $50 *5858 = 292k\text{ GFLOPs}$
> - *CoME*: 50 forward steps:  $50 * (150 + 600 + 206 + 585)\text{ GFLOPs} *2 = 144k\text{GFLOPs}$
> 	 - $+ (100 \text{ Backward Steps}*(419+104)\text{ GFLOPs}=52k \text{ GFLOPs}$
>
> | **config** | **total**   | **trainable** | **GFLOPs(fwd)** | **GFLOPs(bwd)** |
> | ---------- | ----------- | ------------- | --------------- | --------------- |
> | Base       | 57,918,992  | 0             | 585.89          | n/a             |
> | STM        | 32,662,720  | 0             | 206.32          | n/a             |
> | LTM        | 59,500,304  | 1,581,312     | 602.83          | 419.45          |
> | SLTM       | 59,550,272  | 1,582,080     | 150.92          | 104.92          |
> |            |             |               |                 |                 |
> | SSM        | 58,114,832  | 0             | 6,293.09        | n/a             |
> | Sliding    | 57,918,992  | 0             | 11,198.53       | n/a             |
> | Full       | 57,918,992  | 0             | 5,858.95        | n/a             |

---

### Official Review · Reviewer_LCCy · 2025-10-31

**Soundness:** 3
**Presentation:** 2
**Contribution:** 3
**Rating:** 4
**Confidence:** 2

**Summary:**

- The authors propose approaches for augmenting diffusion world models with external memories (a short, long, and short-long term memory) that enable diffusion world models to have better long-term rollouts. The authors propose a product of contrastive experts approach that improves performance, and demonstrate broadly that their proposed approach has good properties empirically and loosely theoretically.

**Strengths:**

- The results are generally very strong, with the proposed CoME generally outperforming all existing models in LPIPS, SSIM, and PSNR.
- The writing and motivation are good, as I agree that incorporating short/long/short+long term memory into existing models would be very impactful.
- The ablations are generally very strong in terms of supporting the author's claims.

**Weaknesses:**

- It's not clear that a fair comparison was done to me, which is my biggest concern. The authors don't include rich details regarding the parameters/FLOPs of models in Table 1, Table 2, and table 3. This makes it hard to determine whether or not the memories are working as intended and really providing strong performance for their size, or are just giving the model more capacity.
    - I see this as the largest issue. Is it possible to reveal this information as well as do a comparison between a baseline model with more FLOPs/parameters to match the size of the baseline with the memory experts?
    - Further, is it possible to do comparisons at multiple different model sizes, to see if the approach scales well?
- Visual perceptive metrics are the primarily method for evaluating the model. While this is common, pixel level reconstruction has been shown to be less effective than modeling in a learned latent space [1], meaning the world models may produce higher fidelity rollouts but not actually be better world models. What do the authors think of this?

[1] https://arxiv.org/abs/2506.09985

**Questions:**

- Overall, I'm open to revising my score, but that primarily relies on addressing my concerns/questions given in the weaknesses.

---

> ### Author Response · Authors · 2025-11-14
>
> We appreciate the reviewer’s thoughtful and detailed feedback.
>
> ## Weaknesses
>
> **FLOPs / Parameter Comparisons** We agree that transparent compute comparisons are essential. Below, we provide full FLOPs and parameter counts for all configurations across Tables 1–3, and we will include these tables in the camera-ready version.
>
> Across the experiments in Tables 1 and 3, CoME operates at comparable compute cost to the baselines. For NWM (Table 2), the larger STM context window increases total compute by an acceptable margin given the improvements in the performance. We additionally want to highlight a benefit of our approach, that CoME remains fully paralelizable across devices, so inference can be limited only by the slowest expert, not by sequential dependencies.
>
> **Quality vs. Consistency** To isolate the effect of memory from model capacity, we added a parameter-matched baseline (DiT-match). This comparison shows a clear separation:
> - **Base:** 0.209 LPIPS
> - **DiT-match:** 0.191 LPIPS
> - **CoME:** 0.097 LPIPS
> The matched model (170M parameters) improves in LPIPs over the base model (50M parameters) but lacks behind CoME.
>
> Furthermore our downstream planning results (trajectory error Table 2.) support the interpretation that CoME’s gains stem from more stable long-horizon modeling, not from added representational capacity and thus increase the 'world model usefulness'.
>
> To highlight this point further for RE10K, we add a metric, where we compute cosine similarity of DINO feature maps for the final generated frames. Since DINO emphasizes semantic rather than pixel-level similarity. CoME shows a larger improvement in DINO similarity than in LPIPS, consistent with improved semantic consistency.
>
> | Method   | LPIPS ↓   | DINOSIM ↑ |
> | -------- | --------- | --------- |
> | Base     | 0.405     | 0.668     |
> | **CoME** | **0.359** | **0.853** |
> | HG-v     | 0.414     | 0.582     |
> | HG-t    | 0.400     | 0.629     |
>
> **Model Sizes** Additionally to the newly added parameter adjusted model (see above), our experiments span a wide range of model sizes and architectures, from 8M up to ~700M-parameter navigation world models, showing that CoME is applicable effectively across different sizes. We will clarify this more explicitly in the revised manuscript.
>
> ---
>
> ### **Full FLOPs / Parameter Tables**
>
> #### **Table 1 – Memory Maze**
>
> **Back-of-the-envelope calculation:**
> - *Reference:* 50 forward steps dit_full $50 *5858 = 292k\text{ GFLOPs}$
> - *CoME*: 50 forward steps:  $50 * (150 + 600 + 206 + 585)\text{ GFLOPs} *2 = 144k\text{GFLOPs}$
> 	 - $+ (100 \text{ Backward Steps}*(419+104)\text{ GFLOPs}=52k \text{ GFLOPs}$
>
> | **config** | **total**   | **trainable** | **GFLOPs(fwd)** | **GFLOPs(bwd)** |
> | ---------- | ----------- | ------------- | --------------- | --------------- |
> | Base       | 57,918,992  | 0             | 585.89          | n/a             |
> | STM        | 32,662,720  | 0             | 206.32          | n/a             |
> | LTM        | 59,500,304  | 1,581,312     | 602.83          | 419.45          |
> | SLTM       | 59,550,272  | 1,582,080     | 150.92          | 104.92          |
> |            |             |               |                 |                 |
> | SSM        | 58,114,832  | 0             | 6,293.09        | n/a             |
> | Sliding    | 57,918,992  | 0             | 11,198.53       | n/a             |
> | Full       | 57,918,992  | 0             | 5,858.95        | n/a             |
> | DiT(Match) | 172,645,648 | 0             | 1,752.87        | n/a             |
> ### **Table 2 – NWM**
>
> **Back-of-the-envelope calculation:**
> - *Reference:*   50 forward steps: $50 * 1719 \text{ GFLOPs} = 86k \text{ GFLOPs}$
> - *CoME*:  50 forward steps: $50 * (1719 + 1182 + 1405*2)\text{ GFLOPs} = 285k \text{GFLOPs}$
> 	+ +25 backward steps:  $25 * 805 \text{ GFLOPs} = 21k \text{ GFLOPs}$
>
> | **config** | **total**   | **trainable** | **GFLOPs(fwd)** | **GFLOPs(bwd)** |
> | ---------- | ----------- | ------------- | --------------- | --------------- |
> | CDiT-XL    | 675M| 0             | 1,719.87        | n/a             |
> | CDiT-L-LTM | 464M | 6,3M     | 1,182.43        | 805.95          |
> | CDIT-B-STM | 231M| 0             | 1,405.33        | n/a             |
> ### **Table 3 – R10k**
>
> **Back-of-the-envelope calculation:**
> - *Reference*: HG-t (3 CFG scores): $50 * 3 * 2420 \text{ GFLOPs} = 363k \text{ GFLOPs}$
> - *CoME*: 50 forward steps: $50  * (2420 + 2493)\text{ GFLOPs} = 245k\text{ GFLOPs}$
> 	- + 100 backward steps: $100 * 1161\text{ GFLOPs} = 116k \text{ GFLOPs}$
>
> | **config** | **total**   | **trainable** | **GFLOPs(fwd)** | **GFLOPs(bwd)** |
> | ---------- | ----------- | ------------- | --------------- | --------------- |
> | UViT-Base  | 288M | 0             | 2,420.06        | n/a             |
> | UViT-LTM   | 293M| 4,6M     | 2,493.46        | 1,161.93        |
> | UViT-HG-v  | 288M | 0             | 2 * 2,420.06    | n/a             |
> | UViT-HG-t  | 288M| 0             | 3 * 2,420.06    | n/a             |

---

### Official Review · Reviewer_5qnA · 2025-11-01

**Soundness:** 3
**Presentation:** 2
**Contribution:** 3
**Rating:** 4
**Confidence:** 4

**Summary:**

This paper proposes CoME, which effectively combines long-term and short-term memory in Diffusion-based World Models to address memory limitations. Based on the existing limitations and concerns, I recommend **weak reject**. However, I am willing to raise my score if the authors can address these limitations and concerns.

**Strengths:**

* Well-motivated problem: The fundamental trade-off between context length and computational cost in world models is clearly articulated. The human cognition analogy (STM/LTM) provides good intuition.
* Effective and elegant solution: Using test-time finetuning to overcome context length limitations in the DiT architecture is an effective and concise approach.
* Thorough ablations: Table 4 convincingly demonstrates the necessity of the contrastive formulation; Table 5 analyzes LTM capacity trade-offs.

**Weaknesses:**

* Insufficient baselines: Missing comparisons with key related works (WorldMem [1], StateSpaceDiffuser [2]).
* Low computational efficiency.
* CoME uses 4 models vs single models for baselines, is this a fair comparison?
* No failure case analysis.

[1] WORLDMEM: Long-term Consistent World Simulation with Memory

[2] StateSpaceDiffuser: Bringing Long Context to Diffusion World Models

**Questions:**

* How does CoME world model condition on action input?
* Can the authors provide additional experiments on Minecraft, since many existing works experiment on this environment (Dreamer4 [1], Oasis [2], MineWorld [3])? Or compare with other existing World Model methods on standard datasets (StateSpaceDiffuser [4], RLVR-World [5], GameNGen [6])?
* What is the difference between SLTM and LTM besides patch size and conditioning? Do they both use the same test-time finetuning method? Section E.1 states "similar to LTM" but provides no details.
* If the test set does not include relative and absolute position information (i.e., only contains images and action sequences) would this method still be effective?
* Can the authors provide detailed computational efficiency comparisons with baselines, including end-to-end latency and throughput?
* What is the total parameter count of CoME, can author provides parameter count comparison with existing baselines.
* Why does SLTM use doubled patch size?

[1] Training Agents Inside of Scalable World Models

[2] Oasis: A Universe in a Transformer

[3] MineWorld: a Real-Time and Open-Source Interactive World Model on Minecraft

[4] StateSpaceDiffuser: Bringing Long Context to Diffusion World Models

[5] RLVR-World: Training World Models with Reinforcement Learning

[6] Diffusion Models Are Real-Time Game Engines

---

> ### Author Response · Authors · 2025-11-16
>
> We thank the reviewer for the constructive feedback. Below we address each concern in detail.
>
> ## Weaknesses
> **Insufficient baselines:**
> We want to highlight that we do not view CoME as a competitor to existing world models but as a framework that can be layered onto such world models. We agree that demonstrating this would strengthen the work, and we provide an additional experiment addressing exactly this (see question response below).
>
> **Low computational efficiency:**
> We include an extended compute analysis below and in more detail in our response to reviewer *LCCy*. Even with the full CoME setup, efficiency remains within the range of existing baselines while offering stronger performance. We want to highlight one additional advantage of CoME, that is, its full cross-device parallelizability, unlike sequential methods such as model-size or context-window scaling. In principle, CoME’s inference time is bounded only by the slowest expert’s forward pass.
>
> **Four models vs. single-model baselines:**
> The compute analysis shows that scaling compute through multiple experts rather than growing the architecture/context window is a valid and competitive strategy. Full CoME achieves efficiency comparable to baselines while improving quality. See compute analysis below and Table 1.
>
> **Lack of failure-case analysis:**
> In our experiments we observed failures occur mainly when one expert provides information that the remaining experts see as implausible. We observed this when an expert’s generation fidelity was heavily reduced, e.g. after aggressive parameter reduction, resulting in an inability to generate basic structure. This degrades overall output. We will add concrete examples and discussion in the final version.
>
> ## Questions
> **How does the world model condition on action input?**
> All experiments use standard DiT adaptive LayerNorm for timestep and action conditioning. Discrete actions are mapped using fourier embeddings followed by an MLP. Continuous actions skip the fourier step. We then simply add the action and timestep embeddings.
>
> **Additional experiments on Minecraft or comparisons to other world models?**
> CoME is intended as a general framework rather than a standalone world model, whenever diffusion-based world models are used, CoME can be applied. We evaluate CoME across a broad set of architectures and pretrained diffusion models (Tables 2–3) to show its generality. Applying CoME to different existing world models is sensible, but not all are currently feasible.
> Still, we provide an additional experiment where we apply CoME to the Oasis [1] or Diamond [2] setting on the simpler Minecraft Marsh dataset [3]. We train a diffusion model with 3 input context frames predicting 17 future frames. For the STM, we double the patch size for efficiency and use 33 context frames to predict 17 future frames. Other settings follow Section 4.1. Results:
>
> | Model | LPIPS | SSIM  | PSNR  |
> | ----- | ----- | ----- | ----- |
> | Base  | 0.408 | 0.450 | 16.19 |
> | +STM  | 0.375 | 0.549 | 17.03 |
> | +LTM  | 0.389 | 0.552 | 17.28 |
> | CoME  | 0.369 | 0.574 | 17.78 |
>
> These results indicate that CoME could be effectively applied to requested world models such as Oasis on Minecraft. We make sure to add these results into the camera-ready version.
>
> **Difference between SLTM and LTM beyond patch size and conditioning? Do they share test-time finetuning?**
> SLTM is explicitly conditioned on absolute 3D information. To make this possible, we train a small adapter network that maps relative positional cues (keyboard actions) plus visual inputs into an absolute 3D position vector. Both SLTM and LTM use the same test-time finetuning strategy.
>
> **Effectiveness without explicit relative or absolute position information?**
> MemoryMaze and DMLab provide relative action signals (noop, forward, left, right, forward_left, forward_right). If “action sequences” refers to these discrete transitions, our main results already demonstrate the method’s effectiveness in that setting.
>
> **Detailed computational efficiency comparisons (latency, throughput)?**
> We include an expanded compute analysis below. CoME despite using multiple experts achieves efficiency within the range of baselines while delivering improved performance.
>
> **Total parameter count of CoME? Comparison to baselines?**
> We added the parameter counts in the computational analysis in our answer to reviewer *LCCy*.
>
> **Why does SLTM use a doubled patch size?**
> SLTM is meant to emulate a more lightweight model that can handle histories with lower compute, at some cost in fidelity. Doubling the patch size reduces compute accordingly.
>
> [1] Decart et al. (2024), *Oasis: A Universe in a Transformer*. https://oasis-model.github.io/
>
> [2]Yan, W., et al. Temporally consistent transformers for video generation. ICML, pp. 39062– 39098. PMLR, 2023.
>
> [3]Alonso et al. (2024), *Diffusion for World Modeling: Visual Details Matter in Atari*.

---

### Meta-Review · Area_Chair_7k5v · 2026-01-09

**Summary:**

Reviewers were generally positive about the paper’s idea of composing heterogeneous memory experts for diffusion world models and found the empirical gains convincing. The main concerns shaping the decision were the fairness of comparisons given the use of multiple experts and increased compute (FLOPs/parameters/runtime), the coverage of key baselines and environments (e.g., existing diffusion world models and Minecraft), and the need for clearer discussion of failure cases, stability under expert disagreement, and the intended roles of STM and SLTM.

**Reviewer Concerns:**

Most major concerns were addressed in the rebuttal. For (LCCy), the authors added detailed FLOPs and parameter tables, a parameter-matched baseline, and semantic/planning-based evaluations, substantially reducing the capacity-confound concern. For (5qnA), additional experiments, clearer action conditioning, and clarification of LTM vs. SLTM resolved most requests, though coverage of some related baselines remains limited. For (9rob) and (hHXL), the authors clarified failure modes, stability, and limitations, but more exhaustive analysis of expert disagreement and role-specific evaluations of STM/SLTM remain open directions.

**Reviewer Scores:**

Reviewer (LCCy) would likely increase their score after the addition of transparent compute reporting, a parameter-matched baseline, and semantic/planning-based evaluations.

Reviewer (5qnA) would plausibly raise their score, given the added compute analysis, architectural clarifications, and the new Minecraft experiment, despite some remaining baseline gaps.

Reviewer (DBh7) would likely maintain a strong accept-level score (around 8), as their main concerns about baselines and computational cost were directly addressed.

Reviewer (9rob) would likely maintain or slightly strengthen an accept-level score (around 8–9), as their concerns regarding compute trade-offs, limitations, and the roles of STM and SLTM were largely resolved.

Reviewer (hHXL) would likely increase their score modestly from 6 to around 6–7, given the added runtime analysis and stability discussion, while some broader modeling assumptions remain open.

---

### Decision · Program_Chairs · 2026-01-26

Accept (Poster)